# Distinct patterns of brain activity mediate perceptual and motor and autonomic responses to noxious stimuli

Laura Tiemann [1], Vanessa D. Hohn [1], Son Ta Dinh[1], Elisabeth S. May[1], Moritz M. Nickel [1], Joachim Gross[2,3] & Markus Ploner [1]

Pain is a complex phenomenon involving perceptual, motor, and autonomic responses, but how the brain translates noxious stimuli into these different dimensions of pain is unclear. Here, we assessed perceptual, motor, and autonomic responses to brief noxious heat stimuli and recorded brain activity using electroencephalography (EEG) in humans. Multilevel mediation analysis reveals that each pain dimension is subserved by a distinct pattern of EEG responses and, conversely, that each EEG response differentially contributes to the different dimensions of pain. In particular, the translation of noxious stimuli into autonomic and motor responses involved the earliest N1 wave, whereas pain perception was mediated by later N2 and P2 waves. Gamma oscillations mediated motor responses rather than pain perception. These findings represent progress towards a mechanistic understanding of the brain processes translating noxious stimuli into pain and suggest that perceptual, motor, and autonomic dimensions of pain are partially independent rather than serial processes.

[1] Department of Neurology and TUM-Neuroimaging Center, Technische Universität München, Ismaninger Str. 22, 81675 Munich, Germany. [2] Institute for Biomagnetism and Biosignalanalysis, University of Münster, Malmedyweg 15, 48149 Münster, Germany. [3] Centre for Cognitive Neuroimaging, University of Glasgow, 62 Hillhead Street, Glasgow G12 8QB, UK. Correspondence and requests for materials should be addressed to M.P. (email: markus.ploner@tum.de)

  1

Pain is commonly defined as an unpleasant sensory and emotional experience associated with actual or potential tissue damage[1] and has, thus, mostly been conceptualized as a perceptual phenomenon. However, the crucial protective function of pain depends on motor responses rather than on perception. Moreover, such motor responses need energy resources, which have to be allocated by the autonomic nervous system. Pain, thus, essentially comprises perceptual, motor, and autonomic dimensions[2], and the protective function of pain eventually depends on the successful translation of noxious stimuli into these dimensions.

The brain mechanisms underlying the translation of noxious stimuli into the different dimensions of pain are not fully known yet. Functional imaging studies have revealed that pain is associated with activation of an extended network of brain regions, including sensory, motor, cingulate, insular, and prefrontal cortices as well as subcortical areas[3,4]. Neurophysiological recordings have specified that noxious stimuli yield a sequence of evoked potentials, including responses termed N1, N2, and P2 waves[5,6]. In addition, they have disclosed that noxious stimuli suppress neuronal oscillations at alpha (8–13 Hz) and beta (13–28 Hz) frequencies and induce oscillations at gamma (30–100 Hz) frequencies[7]. So far, the functional significance of these brain responses has mostly been assessed by analyzing bivariate relationships between noxious stimuli and brain activity, or between brain activity and pain perception. These analyses have shown that the sequence of N1, N2, and P2 waves represents a gradual progress from processes reflecting noxious stimulus characteristics to processes directly or indirectly related to the perception of pain[8,9]. It has further been shown that gamma oscillations induced by noxious stimuli provide complementary information, which is often[10–12] but not always[10,13] closely related to the perception of pain.

A recent series of seminal studies[14–17] extended these bivariate approaches by using mediation analysis[18]. Mediation analysis is a statistical approach, which not only assesses whether an independent variable causes a change in a dependent variable, but investigates how a third variable termed mediator contributes to this change. Mediation analysis can therefore quantify how brain activity is involved in the translation of noxious stimuli and/or psychological interventions into pain. Applications of mediation analysis to functional imaging data[14–17] have revealed that distinct spatial patterns of brain activity mediate the effects of stimulus intensity and psychological interventions on the perception of pain. However, an approach that directly assesses and compares how the brain transforms noxious stimulus information into perceptual, motor, and autonomic responses is lacking so far. Understanding these translation processes promises novel insights into the brain mechanisms of pain. Moreover, their dynamics have conceptual implications for the understanding of pain. Sequential translation processes, e.g., with early brain responses mediating perception and later brain responses mediating motor and autonomic dimensions would indicate a serial organization in which motor and autonomic responses depend on perceptual processes. Alternatively, involvement of early brain responses in the translation into motor and/or autonomic dimensions of pain would indicate a rather parallel organization, in which motor and autonomic responses are partially independent from perceptual processes.

Here we assessed perceptual, motor, and autonomic responses to brief noxious stimuli while recording brain activity using electroencephalography (EEG) in healthy human participants. To assess how EEG responses at different latencies and frequencies translate noxious stimuli into perceptual, motor, and autonomic responses, we performed multilevel mediation analysis[18]. This approach builds upon previous applications of mediation analysis

to functional imaging studies[14–17] and extends them by investigating different dimensions of pain and differentiating between brain responses at different latencies and frequencies. Our findings reveal that distinct patterns of brain responses are involved in the translation of noxious stimuli into the different dimensions of pain. Involvement of the earliest brain responses in mediating motor and autonomic responses suggest a concept of pain in which perceptual, motor, and autonomic responses are partially independent rather than serial processes.

## Results
**Experiment.** In 51 healthy participants, we investigated how the brain translates noxious stimuli into the perceptual, motor, and autonomic dimensions of pain. The experiment comprised three core conditions (Fig. 1). In each condition, 60 brief painful laser

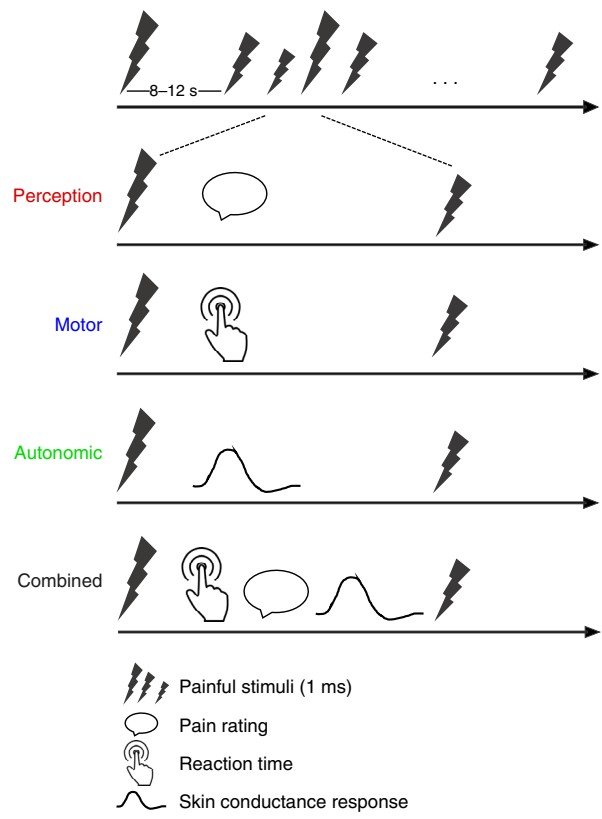

**Fig. 1** Paradigm. The paradigm comprised three core conditions (perception, motor, and autonomic) and an additional combined condition, which were presented in pseudorandomized order. In each condition, 60 painful laser stimuli were applied to the dorsum of the left hand. Stimulus intensity was varied in a pseudorandomized sequence between three individually adjusted levels (low, medium, and high). The interstimulus interval was varied between 8 and 12 s. In the perception condition, participants were prompted to verbally rate the perceived pain intensity on a numerical rating scale (0–100). Pain ratings served as a measure of the perceptual dimension of pain. In the motor condition, participants were instructed to release a button pressed with the index finger of the right hand as fast as possible in response to noxious stimuli. Reaction times served as a measure of the motor dimension of pain. In the autonomic condition, participants were instructed to focus on the painful stimulation without any particular task while skin conductance responses (SCRs) were recorded. SCRs served as a measure of the autonomic dimension of pain. In the combined condition, the participants were asked to first release the button as fast as possible in response to the noxious stimulus and then provide a pain rating. In addition, SCRs were recorded

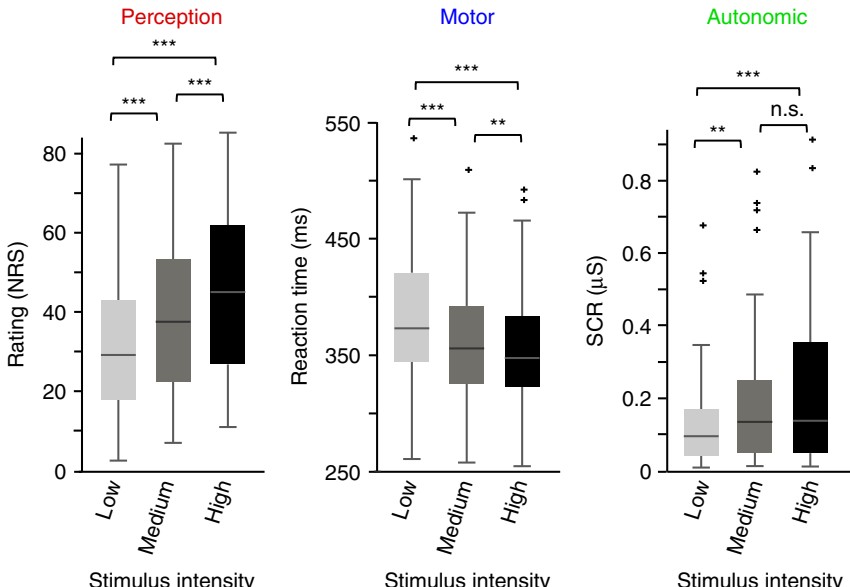

**Fig. 2** Perceptual, motor, and autonomic responses to noxious stimuli. Box plots of pain ratings (0–100, NRS), reaction times (ms), and skin conductance responses (µS) to noxious stimuli of low, medium, and high intensity in the perception, motor, and autonomic conditions, respectively. The band inside the box indicates the median, and the bottom and top edges of the box indicate the 25th and 75th percentiles, respectively. The whiskers extend to the most extreme data points not considered outliers, and the outliers are plotted individually using the "+" symbol. **$p < 0.01$, ***$p < 0.001$, n.s. not significant, dependent samples $t$-tests; NRS numerical rating scale, SCR skin conductance response

stimuli were applied to the left hand. These stimuli selectively activate nociceptive nerve fibers[19] and therefore allow for investigating the translation of nociceptive information into pain without confounding tactile stimulation. Stimulus intensity was varied between three individually adjusted levels (low, medium, and high). In the perception condition, participants were prompted by an auditory cue to verbally rate the perceived pain intensity on a numerical rating scale (NRS) ranging from 0 (no pain) to 100 (worst tolerable pain). Pain ratings served as a measure of the perceptual dimension of pain. In the motor condition, participants were instructed to release a button with the index finger of the right hand as fast as possible in response to the painful stimuli. Reaction times served as a measure of the motor dimension of pain. During the autonomic condition, participants were instructed to focus on the painful stimulation without any further task while skin conductance responses (SCRs) were recorded. SCRs served as a measure of the autonomic dimension of pain. During all conditions, brain activity was recorded using EEG. In addition, we acquired data in a fourth combined condition in which pain ratings, reaction times, and SCRs were not recorded separately but together in each trial.

**Perceptual, motor, and autonomic responses to noxious stimuli.** To investigate whether our experimental manipulation, i.e. the variation of laser intensity, had the expected effects, we first determined whether noxious stimulus intensity influenced perceptual, motor, and autonomic responses. One-way repeated-measures analyses of variance (ANOVAs) showed that noxious stimulus intensity significantly influenced pain ratings, reaction times and SCRs (Fig. 2; perception: $F_{(1, 62)} = 75.54$, $p < 0.001$; motor: $F_{(2, 100)} = 54.20$, $p < 0.001$; autonomic: $F_{(2, 64)} = 10.76$, $p < 0.001$). Pain ratings (low: $34 \pm 21$, medium: $40 \pm 20$, high: $46 \pm 20$; mean $\pm$ SD) and SCRs (low: $0.15 \pm 0.17$ µS, medium: $0.21 \pm .23$ µS, high: $0.23 \pm 0.24$ µS) increased with increasing stimulus intensity, whereas reaction times decreased (low: $375 \pm 78$ ms, medium: $355 \pm 73$ ms, high: $346 \pm 71$ ms). Post hoc pairwise comparisons confirmed that all outcome measures differed

significantly between stimulus intensities (dependent samples $t$-tests, all $p < 0.02$) with the exception of a nonsignificant difference between the SCRs at medium and high stimulus intensities ($t_{(32)} = -1.36$, $p = 0.54$). Thus, as expected, increasing noxious stimulus intensity was associated with higher pain ratings, faster reactions, and stronger autonomic responses. Perceptual, motor, and autonomic responses to noxious stimuli in the combined condition are shown in Supplementary Fig. 1.

**Brain responses to noxious stimuli.** We next assessed brain responses to noxious stimuli in the time and time-frequency domain. Time-domain analysis confirmed that noxious stimuli yielded evoked potentials with the well-known sequence of N1, N2, and P2 waves at latencies around 160, 190, and 300 ms, respectively (Fig. 3, Table 1, Supplementary Fig. 2 and 3)[5,6]. Time-frequency analysis showed that noxious stimuli evoked increases of neuronal oscillations at frequencies below 10 Hz and latencies between 100 and 400 ms, which reflect the evoked potentials[11] and are captured by the time-domain analysis. Moreover, noxious stimuli induced increases of neuronal oscillations at gamma frequencies with peak latencies around 230 ms (Fig. 3, Table 1, Supplementary Fig. 2 and 3)[7]. In addition, noxious stimuli induced suppressions of neuronal oscillations at alpha and beta frequencies[7]. However, these responses occurred after the button releases and can therefore not contribute to the translation of noxious stimuli into motor responses. Thus, they were not further analyzed. Amplitudes of all analyzed responses were significant in comparison to a 1 s prestimulus baseline (dependent samples $t$-tests; $p < 0.01$ for all comparisons). Moreover, amplitudes were modulated by stimulus intensity as indicated by 3 (intensity levels) × 3 (conditions) repeated-measures ANOVAs. As expected, amplitudes of all responses either increased (P2 and gamma) or decreased (N1 and N2) with increasing stimulus intensity (N1: $F_{(2, 80)} = 13.66$, $p < 0.001$; N2: $F_{(2, 76)} = 6.30$, $p = 0.003$; P2: $F_{(2, 82)} = 30.26$, $p < 0.001$; gamma: $F_{(2, 98)} = 13.78$, $p < 0.001$; Greenhouse-Geisser corrected where necessary). In addition, amplitudes of N1 ($F_{(2, 98)} = 4.89$,

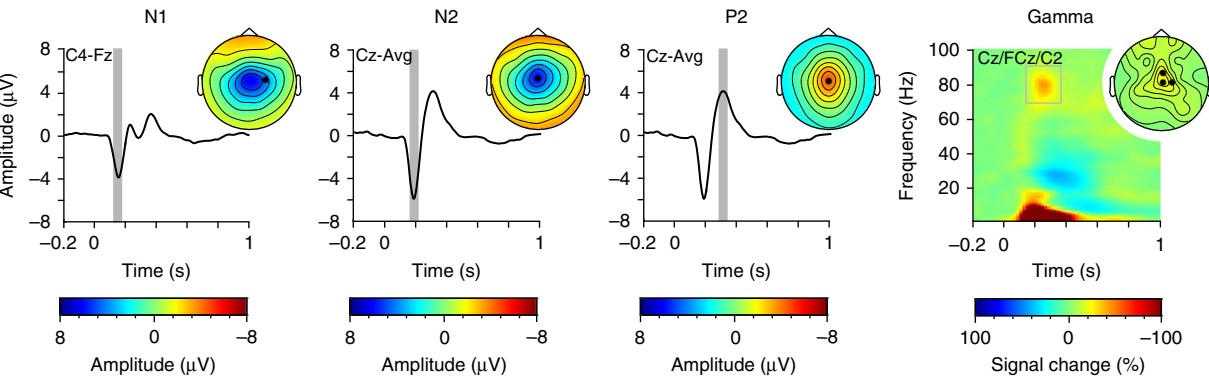

**Fig. 3** Brain responses to noxious stimuli. Mean time courses and time-frequency representation (TFR, right panel) of brain responses averaged across conditions and participants. Marked time periods and time-frequency windows indicate periods/windows chosen to quantify N1, N2, P2, and gamma responses. Topographies depict the scalp distribution of neural activity in these periods/windows, electrodes used for the quantification of the different responses are marked. For visualization only, the TFR is displayed as %-signal change relative to a prestimulus baseline (−1000 to 0 ms)

### Table 1 Peak latencies [ms] of brain responses (mean ± SD)

|  | N1 | N2 | P2 | Gamma |
|---|---|---|---|---|
| Perception | 164 ± 6 | 194 ± 7 | 306 ± 7 | 228 ± 41 |
| Motor | 160 ± 6 | 192 ± 8 | 310 ± 6 | 240 ± 50 |
| Autonomic | 163 ± 7 | 193 ± 7 | 304 ± 6 | 234 ± 36 |

To determine mean peak latencies for the N1, N2, P2 wave, and gamma oscillations, EEG data were first averaged across trials. Then, the latency of the individual peak amplitude in the respective preselected time-electrode(-frequency) windows was determined. Finally, individual peak latencies were averaged across subjects

$p < 0.015$) and gamma ($F_{(1, 66)} = 7.21$, $p = 0.001$) but not of N2 ($F_{(2, 87)} = 1.97$, $p = 0.15$) and P2 ($F_{(2, 98)} = 0.12$, $p = 0.90$) responses were influenced by condition. Post hoc pairwise comparisons confirmed a significantly more negative N1 response amplitude in the motor than in the perception ($t_{(49)} = 4.37$, $p < 0.001$) and autonomic conditions ($t_{(49)} = -4.29$, $p < 0.001$) as well as stronger gamma responses in the motor than in the autonomic condition ($t_{(49)} = 3.02$, $p = 0.01$; all $p$-values Bonferroni-corrected). Taken together, noxious stimuli elicited a well-known pattern of electrophysiological responses, including N1, N2, and P2 waves[5,6], and gamma oscillations[7], which were influenced by stimulus intensity and in part by condition.

**Brain mediators of perceptual, motor, and autonomic dimensions of pain.** To investigate how the different brain responses translate noxious stimuli into the perceptual, motor, and autonomic dimensions of pain, we performed multilevel mediation analyses[18]. Mediation analysis quantifies how a variable termed mediator influences the effects of an independent variable on a dependent variable (Fig. 4a). An extension of the approach termed three-path mediation analyses quantifies the effects of two sequential mediators (Fig. 4b). Mediation analysis is ideally suited to investigate stimulus-brain-outcome relationships and has increasingly been applied to neuroimaging data during recent years[14–17,20]. In all mediation models of the present study, noxious stimulus intensity was the independent variable ($X$). Depending on the condition, pain ratings, reaction times, or SCRs were the dependent variables ($Y$). Single-trial brain responses were the mediators ($M$). Separate mediation models were calculated for each brain response and condition.

**Mediation effects of N1, N2, P2, and gamma responses.** We first investigated how the N1, N2, P2, and gamma responses

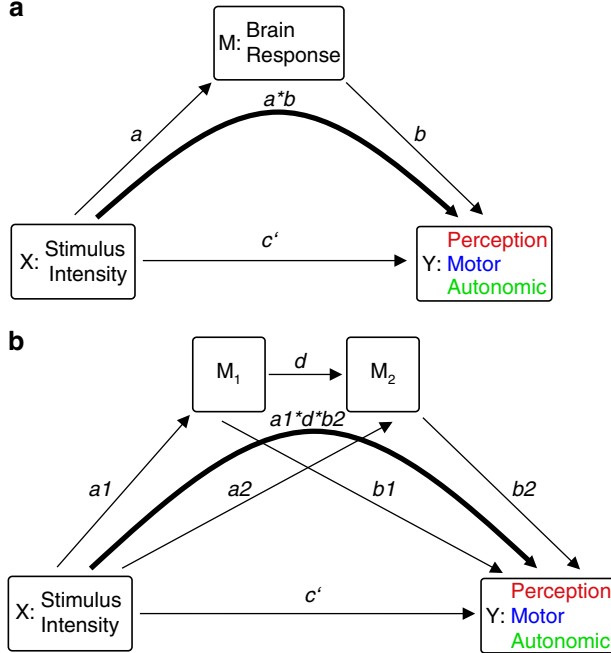

**Fig. 4** Mediation analysis. **a** Two-path mediation model with a representing the relation of $X$ to $M$, b the relation of $M$ to $Y$ controlled for $X$, and c′ the relation of $X$ to $Y$ controlled for $M$. Mediation effects are calculated by multiplying coefficients of path a and path b and tested for significance using a bootstrap approach. **b** Three-path mediation model linking stimulus intensity ($X$) and the perceptual, motor, or autonomic dimension of pain ($Y$) via two sequential mediators ($M1$ and $M2$)

translate noxious stimuli into the different pain dimensions. To this end, we performed mediation analyses with N1, N2, P2, and gamma responses as mediators based on preselected electrodes and time(-frequency) windows (Fig. 3). Figure 5 shows the mediation effects for N1, N2, P2, and gamma responses for each condition and Table 2 shows all second-level path coefficients. In the perception condition, the N2 and P2 waves significantly mediated the effect of stimulus intensity on pain ratings (N2: $\beta_{ab} = 0.003$, $p = 0.01$; P2: $\beta_{ab} = 0.004$, $p = 0.02$). In the motor condition, N1, P2, and gamma responses significantly mediated the effect of stimulus intensity on reaction times (N1: $\beta_{ab} = -0.004$, $p = 0.02$; P2: $\beta_{ab} = -0.005$, $p = 0.02$; gamma: $\beta_{ab} = -0.003$, $p = 0.02$). In the autonomic condition, only the

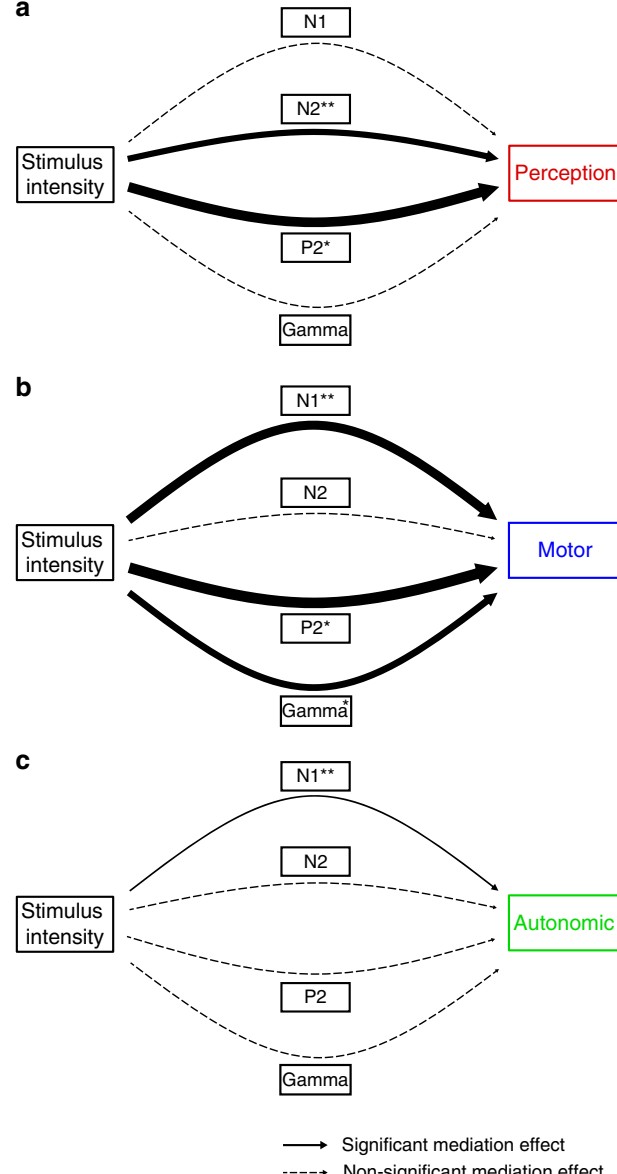

**Fig. 5** Brain mediators of perceptual, motor, and autonomic responses to noxious stimuli. Mediation effects in the perception (**a**), motor (**b**), and autonomic (**c**) conditions. The thickness of the arrows reflects the size of the regression coefficients and thus, represents the strength of mediation effects. Significant mediation effects are indicated by continuous arrows

N1 wave significantly mediated the effect of stimulus intensity on SCRs (N1: $\beta_{ab} = 0.007$, $p = 0.01$). The results, thus, reveal that the perceptual, motor, and autonomic dimensions of pain are mediated by distinct patterns of brain responses. Conversely, each brain response differentially contributes to the different dimensions of pain. The N1 wave mediates autonomic and motor dimensions, the N2 and P2 waves the motor and perceptual dimensions, and gamma oscillations the motor dimension of pain.

In order to assess whether other than the predefined responses mediate between stimulus intensity and the different pain dimensions, additional mediation analyses for the whole time, frequency, and electrode space were performed. These analyses showed clusters, which were spatially, temporally, and spectrally centered around the preselected N1, N2, P2, and gamma responses (Supplementary Fig. 4–6). Hence, we did not observe

mediation effects which were not captured by the preselected responses.

Next, we performed mediation analyses with N1, N2, P2, and gamma responses as mediators for the combined condition. Supplementary Table 1 shows all second-level path coefficients. The results showed that the pattern of path a and path b effects in the combined condition was similar to the perception, motor, and autonomic conditions. The mediation effects, on the other hand, were not statistically significant (all $p > 0.1$, false discovery rate (FDR)-corrected), yet the overall pattern of mediation effects was similar to the perception, motor, and autonomic conditions.

**Comparison of mediation patterns across conditions**. In a next step, we aimed to directly compare the patterns of mediation effects across conditions. To allow for these comparisons, we standardized the mediation effects of brain responses. More specifically, we determined the relative mediation effect of each brain response in comparison to the sum of mediation effects of all brain responses for each condition. A two-way repeated-measures ANOVA with condition (three levels: perception, motor, and autonomic) and brain response (four levels: N1, N2, P2, and gamma) as factors showed a significant interaction effect ($F_{(4, 115)} = 9.89$, $p < 0.001$), indicating that the pattern of relative mediation effects of brain responses differs between conditions. Figure 6 visualizes the patterns of relative mediation effects across brain responses and conditions. The radar chart illustrates that each pain dimension is served by a distinct pattern of brain responses and, conversely, that each brain response differentially contributes to the different dimensions of pain.

**Consistency, uniqueness, and redundancy of mediation effects**. Next, we determined whether the mediation effects were consistent across participants. In multilevel mediation analysis, a significant second-level mediation effect can be driven by effects which are consistent across participants. Such interindividual consistency is indicated by an absence of a significant covariance between path a and b coefficients[21]. Alternatively, significant second-level mediation can be driven by first-level mediation effects which are significant, but differ between subjects regarding their strength and/or direction. Such interindividual variability in mediation is indicated by a significant covariance between path a and b coefficients. We therefore assessed the covariance of path a and b coefficients by calculating Pearson correlations. With the exception of a significant correlation for the N1 response in the motor condition ($r = -0.37$, $p = 0.009$), the analyses did not show significant correlations between path a and b coefficients (perception: N2: $r = 0.08$, $p = 0.6$, P2: $r = -0.05$, $p = 0.7$; motor: gamma: $r = -0.02$, $p = 0.9$, P2: $r = -0.22$, $p = 0.1$; autonomic: $r = 0.09$, $p = 0.6$). Thus, the observed mediation effects were mostly consistent across participants.

We further asked whether the different brain responses provided redundant or complementary information about the translation of noxious stimuli into a certain pain dimension. Thus, for each condition, we performed additional mediation analyses for each brain response in which the remaining brain responses, previously identified as significant mediator(s), were included as covariate(s). These analyses showed that the N2 and P2 waves remained significant mediators between stimulus intensity and the perceptual dimension of pain when controlling for one another (N2, $\beta_{ab} = 0.003$, $p = 0.005$; P2, $\beta_{ab} = 0.004$, $p = 0.001$). Similarly, in the motor condition, N1, P2, and gamma response remained significant mediators when controlling for the other brain responses (N1, $\beta_{ab} = -0.003$, $p = 0.047$; P2, $\beta_{ab} = -0.004$, $p = 0.033$; gamma, $\beta_{ab} = -0.003$, $p = 0.031$). Thus, the different brain responses provide unique and complementary

**Table 2 Results of the two-path mediation analysis**

| | Perception | | | | Motor | | | | Autonomic | | | |
|---|---|---|---|---|---|---|---|---|---|---|---|---|
| | β | SE | Z | p | β | SE | Z | p | β | SE | Z | p |
| **N1** | | | | | | | | | | | | |
| a | −0.0237 | 0.0162 | −1.532 | 0.1256 | −0.0235 | 0.0179 | −1.261 | 0.2763 | −0.0242 | 0.0219 | −1.076 | 0.3092 |
| b | −0.0843 | 0.0143 | −3.646 | **0.0005** | 0.1179 | 0.0228 | 3.815 | **0.0003** | −0.0242 | 0.0085 | −2.816 | **0.0049** |
| c′ | 0.1757 | 0.0174 | 3.928 | **0.0001** | −0.1331 | 0.0176 | −3.486 | **0.0005** | 0.0609 | 0.0132 | 4.139 | **0.0000** |
| c | 0.1804 | 0.0172 | 3.867 | **0.0001** | −0.1439 | 0.0176 | −3.468 | **0.0005** | 0.0628 | 0.0128 | 4.088 | **0.0000** |
| a × b | 0.0012 | 0.0009 | 1.234 | 0.2172 | −0.0038 | 0.0014 | −2.778 | **0.0157** | 0.0007 | 0.0004 | 3.036 | **0.0096** |
| **N2** | | | | | | | | | | | | |
| a | −0.0481 | 0.0147 | −3.171 | **0.0020** | −0.0102 | 0.0172 | −.5843 | 0.5590 | −0.0227 | 0.0223 | −1.017 | 0.3092 |
| b | −0.1033 | 0.0184 | −3.384 | **0.0007** | 0.1652 | 0.0301 | 3.717 | **0.0003** | −0.0677 | 0.0183 | −3.491 | **0.0006** |
| c′ | 0.1736 | 0.0172 | 3.939 | **0.0001** | −0.1393 | 0.0185 | −3.505 | **0.0005** | 0.0598 | 0.0118 | 4.071 | **0.0000** |
| c | 0.1804 | 0.0175 | 3.937 | **0.0001** | −0.1439 | 0.0175 | −3.451 | **0.0005** | 0.0628 | 0.0127 | 4.063 | **0.0000** |
| a × b | 0.0026 | 0.0010 | 2.946 | **0.0129** | 0.0000 | 0.0019 | .0156 | 0.9876 | 0.0008 | 0.0012 | 0.6675 | 0.6343 |
| **P2** | | | | | | | | | | | | |
| a | 0.0965 | 0.0170 | 3.694 | **0.0009** | 0.0811 | 0.0182 | 4.021 | **0.0002** | 0.0417 | 0.0135 | 2.924 | 0.0138 |
| b | 0.1141 | 0.0174 | 3.655 | **0.0005** | −0.0923 | 0.0262 | −3.680 | **0.0003** | 0.0353 | 0.0075 | 3.501 | **0.0006** |
| c′ | 0.1665 | 0.0157 | 3.827 | **0.0001** | −0.1269 | 0.0152 | −3.467 | **0.0005** | 0.0577 | 0.0121 | 4.134 | **0.0000** |
| c | 0.1804 | 0.0174 | 3.894 | **0.0001** | −0.1439 | 0.0171 | −3.429 | **0.0005** | 0.0628 | 0.0129 | 4.066 | **0.0000** |
| a × b | 0.0041 | 0.0016 | 2.533 | **0.0226** | −0.0045 | 0.0017 | −2.658 | **0.0157** | 0.0005 | 0.0003 | 1.366 | 0.3440 |
| **Gamma** | | | | | | | | | | | | |
| a | 0.0250 | 0.0086 | 3.258 | **0.0020** | 0.0340 | 0.0074 | 3.522 | **0.0009** | 0.0215 | 0.0129 | 1.8946 | 0.1163 |
| b | 0.0845 | 0.0211 | 3.544 | **0.0005** | −0.0767 | 0.0333 | −2.409 | **0.0160** | 0.0852 | 0.0195 | 4.0505 | **0.0002** |
| c′ | 0.1747 | 0.0171 | 3.931 | **0.0001** | −0.1348 | 0.0171 | −3.533 | **0.0005** | 0.0599 | 0.0123 | 4.1397 | **0.0000** |
| c | 0.1804 | 0.0175 | 3.966 | **0.0001** | −0.1439 | 0.0174 | −3.465 | **0.0005** | 0.0628 | 0.0128 | 4.1040 | **0.0000** |
| a × b | 0.0009 | 0.0007 | 1.397 | 0.2166 | −0.0029 | 0.0012 | −2.412 | **0.0211** | 0.0002 | 0.0005 | 0.4757 | 0.6343 |

β regression coefficient, SE standard error
Given are the second-level statistics for the mediation analyses of N1, N2, P2, and gamma responses in all three conditions. All p-values are FDR-corrected. Significant effects are marked in bold

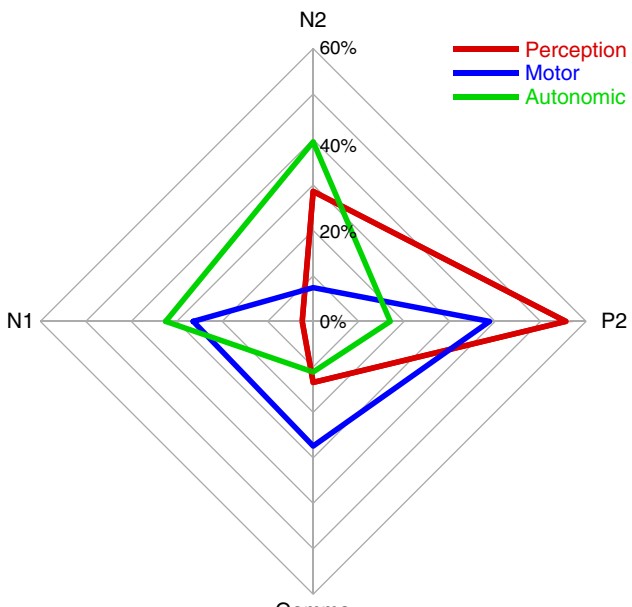

**Fig. 6** Patterns of mediation effects in the different conditions. The radar chart depicts the relative mediation effect for each brain response and condition. The relative mediation effect of a brain response in a certain condition was calculated as the mediation effect divided by the sum of the mediation effects of all brain responses in that condition

information about the translation of noxious stimuli into the different pain dimensions.

**Three-path mediation analyses**. Finally, we investigated whether the different brain responses represent serial mediation steps in

the translation of noxious stimuli into the different pain dimensions. We therefore performed three-path mediation analyses with two sequential mediators (Fig. 4b). In the perception condition, N2 and P2 were included as mediators. In the motor condition, N1 and P2, N1 and gamma, and gamma and P2 were included as mediators. Peak latencies of brain responses (Table 1) determined the sequence in which the two mediators were entered into the model. The results did not show any significant three-path mediation effect in any condition (Fig. 7; all p > 0.5; Supplementary Table 2). Reversing the sequence of the mediators did not yield significant mediation effects either. This lack of serial mediation effects further corroborates that each brain response contributes independently to the translation of noxious stimuli into the different pain dimensions.

**Discussion**

In the present study, we investigated how the brain translates noxious stimuli into the perceptual, motor, and autonomic dimensions of pain. To address this question, we applied multi-level mediation analysis to EEG data in the time and time-frequency domain. The results show that each pain dimension is served by a distinct pattern of brain responses and, conversely, that each brain response differentially contributes to the different dimensions of pain. Beyond, involvement of earliest brain responses in the translation into motor and autonomic responses suggests that the motor and autonomic dimensions do not exclusively depend on perceptual processes. The results, thus, provide physiological support for a concept of pain in which the perceptual, motor, and autonomic dimensions of pain are partially independent rather than serial processes.

In our study, we pursued two novel approaches. First, we distinguished between the perceptual, motor, and autonomic dimensions of pain and designed a simple paradigm to differentially assess these dimensions and the underlying brain

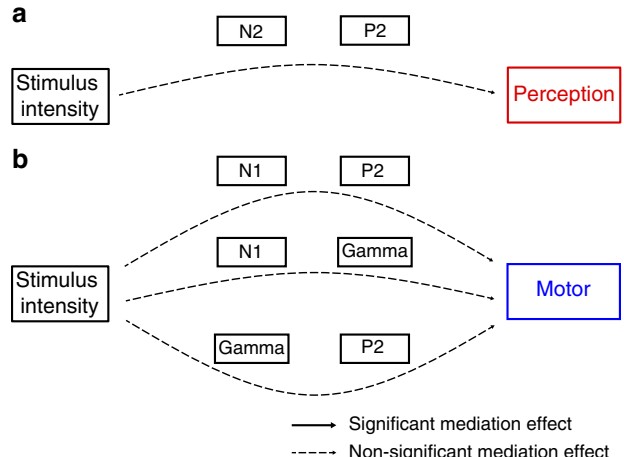

**Fig. 7** Brain mediators of perceptual, motor, and autonomic responses—three-path mediation analysis. Three-path mediation analyses were performed whenever more than one brain response significantly mediated in the two-path mediation models. Thus, three-path mediation analyses were performed for the N2 and P2 waves in the perception condition as well as for the N1 and P2, N1 and gamma, gamma and P2 responses in the motor condition. A schematic overview of potential mediation effects (path a1 × d × b2) in the perception (**a**) and motor (**b**) condition is depicted. Dashed arrows indicate that no significant three-path mediation effects were found

processes. This approach complements and extends the prevailing conceptual and experimental focus on the perceptual dimension of pain. Second, to investigate the brain mechanisms underlying the different pain dimensions, we applied multilevel mediation analysis. Multilevel mediation analysis extends common bivariate analyses between an input and brain activity or between brain activity and an outcome. By quantifying how each brain response is involved in the transformation of noxious stimuli into pain, it represents a step further on the way from correlative to mechanistic insights into stimulus-brain-behavior relationships. Consequently, multilevel mediation analysis has increasingly been applied to neuroimaging data to understand how noxious stimulus intensity and psychological factors translate into pain[14–17,20,22]. Here we have adapted that approach for EEG data. Together, our novel paradigm and multilevel mediation analysis of EEG data allowed for differentially investigating how the brain translates noxious stimuli into the perceptual, motor and autonomic dimensions of pain.

The present results can be interpreted from two perspectives. Regarding the functional significance of brain responses, they show how each EEG response differentially contributes to the different dimensions of pain. Regarding the different dimensions of pain, they reveal that perceptual, motor, and autonomic dimensions of pain are mediated by distinct patterns of EEG responses. In the following, we will discuss these two perspectives together. When discussing the potential generators of EEG responses, it is important to bear in mind that evidence on the neural generators of LEP using source reconstruction is inherently ambiguous.

Our findings show that the N1 wave is particularly involved in translating noxious stimuli into motor and autonomic responses. The N1 mostly originates from primary sensorimotor cortex[23] with additional contributions from operculo-insular and midcingulate/supplementary motor cortices[24]. It reflects an early processing stage, which occurs regardless of whether a noxious stimulus is consciously perceived[8]. The involvement of this wave in the preparation of motor and autonomic responses to noxious

stimuli appears functionally reasonable since these responses must occur rapidly to prevent injury and do not necessarily require conscious stimulus perception. Previous evidence has demonstrated that midcingulate/supplementary motor areas receive direct nociceptive projections[25] and are involved in generating motor as well as autonomic[26] responses. These direct nociceptive projections to motor areas of the brain might subserve the involvement of the N1 wave in translating noxious stimuli into motor and autonomic responses.

Furthermore, we observed that the N2 and P2 waves are involved in translating noxious stimuli into perception and/or motor responses. This finding is well compatible with recent evidence on the functional significance of these responses. Current views claim that these responses essentially reflect the salience of sensory events[27] and that salience, in turn, influences perceptual processes. Moreover, a role of N2 and P2 waves in translating noxious stimuli into pain perception is in line with recent functional magnetic resonance imaging (fMRI) studies[15,17]. These studies have shown that BOLD responses from cingulate and operculo-insular cortices[5,24], which are the main generators of the N2 and P2 waves are involved in translating noxious stimuli into pain perception. Most recent evidence has extended that view by revealing a particularly close link between the N2 and P2 waves, salience, and motor responses[28,29]. The present findings complement and extend these findings by showing that the N2 and P2 waves are directly involved in the translation of noxious stimuli into both perception and motor responses.

Lastly, our results showed that gamma responses, which likely originate from primary somatosensory[11,12] and insular[30] cortex, were significantly involved in translating noxious stimuli into motor responses. At first glance, this observation seems to be at variance with previous evidence showing that gamma responses to noxious stimuli are often[10–12], but not always[10,13], closely related to pain perception. However, in the present study, a significant path b effect in the perception model confirms a significant relationship between gamma responses and pain perception. A significant mediation effect, however, was found for motor responses only. These findings indicate that gamma responses are significantly related to both perceptual and motor responses but mechanistically involved in the translation into motor responses only. This finding is in line with recent evidence demonstrating close relationships between gamma responses and pain behavior in animals[31] as well as reaction times in humans[32]. In this context, the close relationship of gamma responses to pain perception in the present and previous studies might essentially reflect the involvement of gamma responses in the motivation, preparation, and execution of motor responses, which, in turn, are related to pain perception. The present findings therefore suggest a careful reconsideration of the role of gamma oscillations in the processing of pain.

The present findings have further conceptual implications for the understanding of pain. The results revealed that perceptual, motor, and autonomic responses are mediated by unique patterns of brain responses. In particular, they show that the earliest brain responses mediate the motor and autonomic but not the perceptual dimension of pain, which indicates that these dimensions do not exclusively depend on perceptual processes. Instead, our results indicate that the different pain dimensions are at least in part independent processes. This is in line with accumulating evidence in animals[25,33–35] and humans[9,36–38] for a parallel organization of ascending pain pathways projecting to somatosensory, motor, insular, and cingulate cortices as well as to subcortical areas, including amygdala, hypothalamus, and the brainstem. Moreover, the findings are compatible with mutual rather than unidirectional influences between perceptual

and motor processes and, hence, with models emphasizing an important role of motivational[2,39,40] and motor[41–43] processes for pain.

The relationship between perceptual, motor, and autonomic dimensions of pain is also relevant for the understanding of chronic pain. It is increasingly recognized that chronic pain is not only associated with the ongoing perception of pain but also with alterations of motor[41,44] and autonomic[45] processes. The characterization of distinct neural pathways[46] underlying the perceptual, motor, and autonomic dimension of pain might therefore help to understand the complex brain pathology of chronic pain. Moreover, understanding these pathways might help to define their individual abnormalities and to tailor pain treatment. However, it is important to bear in mind that the peripheral[47] and central[3,48] neural mechanisms underlying brief experimental pain processing likely differ from those underlying chronic pain.

Some limitations apply to the present findings and their interpretation. First, an influence of task effects on mediation patterns serving the translation of noxious stimuli into percep-tual, motor, and autonomic responses cannot be ruled out. At first glance, the lack of mediation effects in the combined con-dition suggests such task effects. However, in this condition, influences of motor responses on perceptual[49] and autonomic responses[50] are likely to prevent the separate assessment of the underlying processes. Moreover, task effects on brain responses to noxious stimuli have been shown to manifest as amplitude dif-ferences (e.g. refs. [51–54]). In the present study, amplitude differ-ences between conditions were adjusted prior to the analysis. The only way different tasks could influence mediation weights would therefore be an influence on mediation weights independent from an influence on amplitudes. Moreover, it appears unlikely that different tasks not only modulate, but fundamentally change the processing steps and processing hierarchy which translates a noxious stimulus into a certain response. Second, the specificity for pain remains unclear. fMRI[55] and EEG[56] studies have shown that most, if not all, brain responses to noxious stimuli are not pain-specific but rather reflect the salience of noxious events[27]. It is therefore likely that similar, partially independent patterns of neural responses mediate not only perceptual, motor, and auto-nomic responses to noxious stimuli but also to equally salient and threatening stimuli from other modalities. However, the present findings provide direct evidence for partially independent pro-cesses serving perceptual, motor, and autonomic responses only for noxious stimuli and the particular experimental conditions, e.g. with eyes closed. Third, due to the well-known phenomenon of non-responders[57], the sample size in the autonomic condition was lower than in the other conditions. This might limit the statistical power of that condition. However, our main inter-pretations rest upon differences in the patterns of mediation effects across brain responses and conditions, which are unaffected by sample size. Fourth, EEG has a low sensitivity for brain activity originating from subcortical and brainstem areas, which play an important role in the processing of painful stimuli[3,58]. Hence, methods with a higher sensitivity for deep brain areas might complement the present approach.

In summary, the present study reveals how distinct patterns of brain responses serve the translation of noxious stimuli into the perceptual, motor, and autonomic dimensions of pain. Each EEG response including the N1, N2, P2 waves, and gamma oscillations contributes unique and complementary information to these translation processes. Conversely, each pain dimension is medi-ated by a distinct pattern of brain responses. Moreover, invol-vement of the earliest brain responses in translating noxious stimuli into motor and autonomic but not perceptual responses provides physiological support for a concept of pain in which the

sensory, motor, and autonomic dimensions of pain are partially independent processes. Thus, the present mediation analysis-based approach provides novel mechanistic insights into how the brain subserves different dimensions of pain and bears conceptual implications for the understanding of pain in health and disease. The contribution of task effects to the observed findings and whether they are pain-specific or generalize to other equally salient and threatening events remains to be demonstrated.

## Methods

**Participants.** Fifty-one right-handed healthy participants (25 females) with a mean age of 27 years (range 20–37) participated in the study. Participants were recruited via advertisements on bulletin boards of local universities. Exclusion criteria comprised a history of neurological and psychiatric diseases, including current or recurrent pain as well as the regular use of medication. The study was approved by the local ethics committee and carried out in accordance with the relevant guidelines and regulations. Written informed consent was obtained from each participant.

**Procedure.** To investigate how the brain translates noxious stimuli into perceptual, motor, and autonomic dimensions of pain, the experiment included three core conditions and an additional combined condition, which were presented in ran-domized order (Fig. 1). In each condition, 60 painful stimuli were applied to the dorsum of the left hand. Stimulus intensity was varied between three individually adjusted levels (low [$n = 20$], medium [$n = 20$], and high [$n = 20$]; see below) in a pseudorandomized sequence. Stimuli were presented with an interstimulus interval of 8–12 s, resulting in a duration of approximately 20 min per condition and a total duration of 2.5 h per participant, including preparations and breaks between conditions. In the perception condition, participants were prompted by an auditory cue presented 3 s after each noxious stimulus to verbally rate the perceived pain intensity on a NRS ranging from 0 (no pain) to 100 (worst tolerable pain). Pain ratings served as a measure of the perceptual dimension of pain. In the motor condition, participants were instructed to release a button with the index finger of the right hand as fast as possible in response to each noxious stimulus. Reaction times served as a measure of the motor dimension of pain. During the autonomic condition, participants were instructed to focus on the painful stimulation without any further task while SCRs were recorded. SCRs served as a measure of the autonomic dimension of pain. During all conditions, participants were seated in a comfortable chair with eyes closed and exposed to white noise through headphones to cancel out noise of the laser device.

Preceding the actual experiment, pain thresholds and stimulation intensities were determined (see below) and 10 min of resting state EEG data were recorded, which were not analyzed in the present study. Additionally, a fourth combined condition with a combination of the perceptual and motor tasks was performed. In this condition, the participants were asked to first release the button as fast as possible in response to the noxious stimulus and then provide a pain rating. In addition, SCRs were recorded.

**Stimulation.** Painful stimuli were applied by means of cutaneous laser stimulation, which induces a pinprick-like sensation. Cutaneous laser stimulation selectively activates nociceptive afferents without concomitant activation of tactile afferents and, thus, allows for the selective experimental investigation of pain pathways[19]. The stimuli were applied to the dorsum of the left hand using a Tm:YAG laser (StarMedTec GmbH, Starnberg, Germany) with a wavelength of 1960 nm, a pulse duration of 1 ms, and a spot diameter of 5 mm. In order to avoid tissue damage and habituation/sensitization effects, stimulation sites were slightly changed after each stimulus. A distance pin mounted to the handpiece of the laser device ensured a constant distance of 12 cm between the laser device and the skin surface.

Laser energies were individually adjusted to induce different levels of pain (low, medium, and high). To this end, pain thresholds were determined using the method of limits. Subsequently, 20 stimuli of different supra-threshold laser energies were applied and the induced sensations were rated by the subject on a NRS ranging from 0 (no pain) to 100 (worst tolerable pain). Next, a regression line was fitted to the 20 energy-rating pairs. Laser energies matching individual ratings of NRS 30, 50, and 70 were used for the low-, medium-, and high-intensity stimulation, respectively. Maximum laser energy was 600 mJ. Mean laser energy was 480 ± 40 mJ (mean ± SD) for low-intensity stimulation, 530 ± 40 mJ for medium-intensity stimulation, and 580 ± 50 mJ for high-intensity stimulation. Laser energies differed significantly between low-, medium-, and high-intensity stimulations ($F_{(52, 1)} = 5748.37$, $p < 0.001$, ANOVA; $p < 0.001$ for all post hoc comparisons using dependent-sample $t$-tests).

**Assessment of perceptual, motor, and autonomic dimensions of pain.** Single-trial pain ratings, reaction times, and SCRs served as measures of the perceptual, motor, and autonomic dimensions of pain, respectively. In the perception condi-tion, pain ratings were manually added to the EEG data by the experimenter. Trials in which no rating or a rating of 0 occurred were discarded. In the motor condition, reaction times were measured by a custom-built response box (Brain Products,

Munich, Germany) ensuring an accuracy in millisecond range. Trials in which no or a delayed motor response occurred (reaction times > 650 ms) were discarded to exclude outliers as well as sensations mediated by C-fibers[49,59]. After applying these criteria, an average of 19 out of 20 trials per stimulus intensity (low: 18; medium: 20; high: 20) remained in the perception condition. In the motor condition, an average of 18 out of 20 trials per stimulus intensity (low: 17; medium: 19; high: 19) remained. In the combined condition, an average of 18 out of 20 trials (low: 17; medium: 18; high: 18) remained. Due to technical problems with the recording device data of one subject had to be excluded from further analyses, resulting in a sample size of 50 participants for the motor condition. In the autonomic condition, SCRs were recorded using two Ag/AgCl electrodes, which were attached to the palmar distal phalanges of the left index and middle finger. Data were recorded in direct current mode with a bipolar BrainAmp ExG MR amplifier (Brain Products, Munich, Germany) with a constant voltage of 0.5 V, low-pass filtering at 250 Hz, and a sampling frequency of 1000 Hz. Subsequent offline analysis included low-pass filtering at 1 Hz using a fourth-order Butterworth filter, downsampling to 500 Hz, and a visual artifact inspection of single-trial time traces. For the remaining trials, single-trial SCRs were defined as amplitude difference between the maximal peak and the preceding trough within a search window from 1 to 7.8 s post stimulus following standard peak detection methods[50]. In this condition, 18 participants were excluded from the analysis due to technical problems with the recording device (n = 3) or the absence of SCRs as assessed by visual inspection of single-subject averages (n = 15), resulting in a sample size of 33 participants for the autonomic condition. Single trials were visually inspected for artifacts and discarded when necessary, resulting in a mean trial number of 20 stimuli per stimulus intensity.

**EEG recordings and preprocessing**. EEG data were recorded with an electrode cap (EasyCap, Herrsching, Germany) and BrainAmp MR plus amplifiers (Brain Products, Munich, Germany; input impedances 10 MΩ) using the Brain-Vision Recorder software (Brain Products, Munich, Germany). The electrode montage included 65 scalp electrodes consisting of all electrodes of the International 10–20 system as well as the additional electrodes FPz, AFz, FCz, CPz, POz, Oz, Iz, AF3/4, F5/6, FC1/2/3/4/5/6, FT7/8/9/10, C1/2/5/6, CP1/2/3/4/5/6, P1/2/5/6, TP7/8/9/10, and PO3/4/7/8/9/10. Two additional electrodes were fixed below the outer canthus of each eye. During the recording, the EEG was referenced to the FCz electrode, grounded at AFz, sampled at 1000 Hz, high-pass filtered at 0.015 Hz, and low-pass filtered at 250 Hz. The impedance of all electrodes was kept below 20 kΩ.

EEG data were preprocessed using the BrainVision Analyzer software (Brain Products, Munich, Germany). Data were downsampled to 500 Hz. For artifact detection, a high-pass filter of 1 and a 50 Hz notch filter for line noise removal were applied to the EEG data. Independent component analysis was performed to identify components representing eye movements and muscle artifacts[60]. Furthermore, signals exceeding an amplitude threshold of ±100 μV or displaying a gradient steeper than 30 μV/s were marked for rejection. Next, all data were visually inspected and remaining bad segments marked. Subsequently, independent components representing artifacts were subtracted from the raw, unfiltered EEG data. Previously marked bad segments were removed and EEG data were re-referenced to the average reference. The data were then segmented into 4.5 s epochs ranging from −1500 to 3000 ms with respect to the painful stimulus. After removing trials contaminated by EEG artifacts, an average of 18 out of 20 trials per stimulus intensity remained in the perception (low: 18, range 11–20; medium: 18, range 13–20; high: 18, range 10–20) and the motor condition (low: 16, range 11–20; medium: 18, range 13–20; high: 18, range 10–20). In the autonomic condition, an average of 19 trials remained per stimulus intensity (low: 19, range 16–20; medium: 19, range 16–20; high: 19, range 16–20). In the combined condition, an average of 17 trials remained per stimulus intensity (low: 16, range 3–20; medium: 17, range 3–20; high: 17, range 8–20).

**Time-domain analysis of EEG data**. Subsequent EEG data analyses were performed using FieldTrip[61]. To assess single-trial waveforms of pain-related potentials, data were band-pass filtered from 1 to 30 Hz. No baseline correction was performed in order to enable cluster-based permutation testing against baseline (see below). Amplitudes of the N1, N2, and P2 waves of pain-related potentials were assessed by averaging single-trial amplitude values across predefined time windows[10,28,62–64]. Averaging across time windows was preferred over peak amplitudes as this procedure yields more reliable values in cases of low signal-to-noise ratio inherent to single-trial EEG data. To quantify the N1 wave, data were re-referenced to Fz[65] and averaged across a time window of 150–180 ms at electrode C4. To quantify N2 and P2 waves, mean amplitudes across a time window of 180–210 and 290–320 ms, respectively, were calculated at electrode Cz.

**Time-frequency domain analysis of EEG data**. Prior to transforming raw data from the time to the time-frequency domain, a high-pass filter of 1 Hz and a band-stop filter of 49–51 Hz were applied. Subsequently, a Hanning-tapered fast Fourier transformation was applied using a moving time window with a length of 250 ms and a step size of 20 ms, resulting in single-trial power estimates for the frequencies from 1 to 100 Hz. Responses at gamma frequencies were assessed by averaging single-trial power estimates across a time window of

150–350 ms, a frequency range of 70–90 Hz, and electrodes Cz, FCz, and C2. This time-frequency window was chosen as previous studies showed strongest gamma responses at these latencies and frequencies[10,32,66,67]. Although gamma responses extend to other latencies and frequencies (e.g. ref. [12]), the chosen time-frequency window is therefore likely to capture gamma responses with an optimum signal-to-noise ratio. Brain activity at theta frequencies was not analyzed in the time-frequency domain as these responses represent mainly phase-locked activity[11,12], which is well captured by the time-domain analysis[7,68]. Moreover, later responses at alpha and beta frequencies recorded at latencies > 500 ms[68] were not analyzed as these responses occur after mean reaction times (359 ms + 19) and can therefore not be involved in the translation of noxious stimuli into the motor and perceptual dimensions of pain.

**Multilevel mediation analysis**. To investigate how the brain translates noxious stimuli into the perceptual, motor, and autonomic dimensions of pain, we performed two-path (Fig. 4a) and three-path (Fig. 4b) multilevel mediation analyses as implemented in the Multilevel Mediation and Moderation (M3) Toolbox[69]. Prior to all mediation analyses, the data for X, M, and Y were z-transformed across participants and trials, but not across conditions. By matching mean and standard deviation of the variables across conditions, this procedure corrects for potential amplitude differences between conditions. In all mediation models, noxious stimulus intensity was the independent variable (X). Pain ratings, reaction times, or SCRs were the dependent variable (Y). Single-trial brain activity measures, i.e. N1, N2, P2, or gamma responses, were the mediator (M).

In all two-path mediation models, five path coefficients were calculated for each subject (first-level single-subject coefficients) in a regression-based approach. These five path coefficients quantified the relationship of X to M (path a), the relationship of M to Y controlled for X (path b), the relationship of X to Y (path c), the relationship of X to Y controlled for M (path c′), and the mediation effect (path ab)[70]. Mediation effects are calculated by multiplying coefficients of path a and path b[70]. In addition to these first-level mediation effects, the multilevel mediation analysis incorporates second-level effects. Second-level coefficients represent averaged weighted single-subject coefficients. The respective weights for each subject are determined by means of a weighted least squares-based mixed effects model taking within- and between-subject variance into account. Second-level mediation effects are calculated by multiplying coefficients of path a and path b and adding the covariance of path a and path b[70]. Importantly, in a multilevel mediation model, both path a and path b effects can be significant without significant mediation effect and vice versa. This is due to the fact that mediation analysis aggregates path a and path b effects in one statistically testable term, which represents the translational process between the independent and dependent variable. Thus, the test of mediation (path ab) provides additional information beyond the co-occurrence of significant path a and path b effects[14,71].

Three-path mediation analysis investigates whether two mediators intervene in series between an independent and a dependent variable[72]. Significant mediation is assumed when each of the three relevant paths (a1, d, and b2, Fig. 4b) is significant by itself and the product of the respective path coefficients is significantly different from zero. The multilevel implementation is largely identical to the two-path mediation analysis (for details, see ref. [71]).

**Mediation analyses of N1, N2, P2, and gamma responses**. First, mediation effects of N1, N2, P2, and gamma responses were assessed. To this end, the amplitudes of N1, N2, P2, and gamma responses were entered into separate two-path mediation analyses to examine whether they mediated between stimulus intensity and the different pain dimensions. Thus, four separate mediation analyses were calculated for each condition, resulting in a total of 12 mediation analyses. Additional mediation analyses with covariates were performed whenever more than one brain response was found to be a significant mediator to investigate whether these brain responses carried redundant or complementary information. Likewise, additional three-path mediation analyses were performed using those brain responses, which were found to be significant mediators in the two-path mediation models as first and second mediator. Peak latencies of brain responses (Table 1) determined the sequence in which the two mediators were entered into the model. To control for the plausibility of the results, the three-path mediation models were re-calculated with an inverted sequence of mediators.

Finally, we aimed to compare the pattern of mediation effects across conditions. To this end, for all conditions separately, we determined the relative mediation effect of each brain response in comparison to the total mediation effect of all brain responses combined. The relative mediation effect was calculated based on the following equation (shown here exemplarily for the relative mediation effect of the N2 response in the perceptual condition):

$$\%\mathrm{ME}_{\mathrm{N2Perception}} = \left( \frac{\mathrm{ab}_{\mathrm{N2Perception}}}{\mathrm{ab}_{\mathrm{N1Perception}} + \mathrm{ab}_{\mathrm{N2Perception}} + \mathrm{ab}_{\mathrm{P2Perception}} + \mathrm{ab}_{\mathrm{yPerception}}} \right) \times 100$$

(1)

Importantly, the mediation effect for each brain response was determined based on a mediation model in which all other brain responses were included as covariates.

**Mediation analyses in the whole time, frequency, and electrode space.** To assess whether brain responses not captured by the predefined electrode-time (-frequency) windows mediate between stimulus intensity and the different pain dimensions, we performed additional mediation analyses for every time (-frequency) point and electrode in a time window from −1 to 1 s with respect to the painful stimulus. The multiple-comparison problem was addressed by means of cluster-based non-parametric permutation testing (see Further statistical analysis).

**Further statistical analysis.** To investigate whether noxious stimuli induced intensity-dependent perceptual, motor and autonomic responses, we calculated a one-way repeated-measures ANOVA per condition with intensity as factor (three levels: low, medium, and high) and pain rating, reaction time, and SCR as dependent variables. Whenever the assumption of sphericity was violated, Greenhouse-Geisser corrected values are reported. Significant main effects ($p < 0.05$, two-sided) were followed up by post hoc Bonferroni-corrected dependent samples $t$-tests.

Next, we assessed whether noxious stimuli elicited the established brain responses in the time and time-frequency domain by comparing response amplitudes in a predefined time(frequency) window against average amplitudes in a 1 s prestimulus baseline using dependent samples $t$-tests. In addition, we investigated the effect of stimulus intensity and condition on response amplitudes by calculating a 3 (intensity levels) × 3 (conditions) repeated-measures ANOVA for each brain response. Significant main effects ($p < 0.05$, two-sided) were followed up by post hoc Bonferroni-corrected dependent samples $t$-tests.

Next, we investigated which of the elicited brain responses were involved in the translation from stimulus intensity into the perceptual, motor, and autonomic dimension of pain by means of multilevel mediation analysis (see above). Permutation tests were applied for statistical testing as implemented in the Multilevel Mediation and Moderation (M3) Toolbox[69] (for a detailed description see supplementary Materials in ref. [71]). To correct for multiple comparisons across the four different brain responses, FDR correction[73] was applied.

In case of significant mediation, we tested the covariance of path a and b effects by means of Pearson correlation coefficients. This was done because the strength of the mediation effect in a multilevel mediation analysis is not solely influenced by the product of the means of a and b, but also takes the consistency of effects across participants into account by considering the covariance of path a and b effects (Eq. 9 in ref. [21]). Specifically, the absence of a significant covariance between path a and b coefficients indicates a second-level mediation effect, which is consistent across participants. Significant covariance between path a and b coefficients, on the other hand, indicates that the second-level mediation effect is driven by first-level mediation effects which are significant, but differ in their strength and/or direction.

In order to statistically compare the patterns of mediation effects across conditions, the relative mediation effects for each condition and brain response (see Mediation analyses of N1, N2, P2, and gamma responses) were entered in a two-way repeated-measures ANOVA with condition (three levels: perception, motor, and autonomic) and brain response (four levels: N1, N2, P2, and gamma) as factors. A significant interaction effect would indicate that the relative mediation effects of brain responses differ across conditions, thus indicating that the translation of noxious stimuli into the perceptual, motor, and autonomic dimension of pain is subserved by different patterns of brain responses.

To test for statistical significance and to correct for multiple comparisons in the mediation analyses in the whole time, frequency, and electrode space, we applied cluster-based non-parametric permutation tests as implemented in FieldTrip[74]. This approach effectively corrects the family-wise error rate in the context of multiple comparisons by grouping adjacent data points displaying similar statistical effects and is not affected by partial dependencies in the data[74]. First, a dependent samples $t$-statistic was computed by comparing the single-subject coefficients for path ab at every electrode-time(-frequency) point in the activation period (0 to 1 s post stimulus) versus the corresponding time-averaged baseline period (−1 to 0 s prestimulus). Clusters of neighboring electrodes and time points (and frequencies) with $p < 0.05$ (dependent samples $t$-test) were selected and $t$-values within each cluster were summed up, resulting in cluster-level test statistics. The maximum cluster-level test statistic was then compared to a reference distribution of maximum cluster $t$-value sums obtained by randomly interchanging the labels of the baseline and activation period and recalculating the cluster-level test statistic 1000 times. This comparison resulted in a $p$-value, which was determined by the proportion of permutations in which the maximum cluster-level test statistic exceeded the actually observed maximum cluster-level test statistic in the data. This procedure was repeated for every condition and every brain response. All statistical tests were two-sided with a significance level of 0.05.

**Code availability.** A custom Matlab-code used in the manuscript can be provided upon reasonable request.

## Data availability
The data are available at the OSF online repository [https://osf.io/bsv86/]. A reporting summary for this Article is available as a Supplementary Information file.

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

## Acknowledgements

The study was supported by the Deutsche Forschungsgemeinschaft (PL 321/11–1, PL 321/11-2, and PL 321/13-1), the Bavarian State Ministry of Education, Science and the Arts, and the Wellcome Trust (098433). We thank the Cognitive and Affective Neuroscience Lab, University of Colorado at Boulder for the Multilevel Mediation and Moderation (M3) Toolbox.

## Author contributions

L.T. and M.P. conceived the study; L.T. and S.T.D. acquired the data; L.T., V.D.H., and M. P. analyzed the data, interpreted the results, and wrote the manuscript. All authors contributed to the interpretation of the results and the revision of the manuscript.

**Additional information**

**Competing interests:** The authors declare no competing interests.

