## [Peer Review File · Nature Communications]

Reviewers' comments:

Reviewer #1 (Remarks to the Author):

The submitted study analysed the associations between individual components of LEPs (N1, N2, P2 and gamma band response) in three tasks designed to capture pain intensity, reaction time and autonomic responses. Laser stimuli of three different intensities were used to elicit responses of monotonically increasing amplitudes, and a complex mediation analysis was employed to attribute subsets of LEP responses to either motor, autonomic or subjective responses. Results show that the motor responses were mediated primarily by the gamma responses, subjective responses by the N2 and P2 LEP components, and autonomic responses by the N1 LEP component. Results are novel in teasing apart different types of responses to noxious stimuli and their cortical mediators. The study is well-conducted and important as it sheds light on the functional significance of LEP components. However, there are few points needing amendments and/or explanations which limits reviewer's possibility to comment on a wider impact of the study.

1. The three experiments have been conducted in blocks each requiring the participants to carry out a different task. Therefore, factors such as motor readiness or focusing attention towards the source of pain (rather than on e.g., motor response) might have affected the mediation weights. For instance, the N1 component in LEPs occurring during periods of a high motor response expectancy shows an increased amplitude compared to periods with low level of motor response expectancy (Stancak et al., *Behav. Brain Res.*, 2012). Similarly, focusing attention on the hand receiving a noxious stimulus, a condition perhaps similar to the "autonomic" condition in the present study, has been shown to reduce the N2/P2 components (Longo et al., *J. Neurosci.*, 2009). Could the differences in mediation weights among the three conditions be attributed to the fact that they related to different tasks and instruction sets?

2. Along the same vein as (1), did LEPs differ between the three conditions? While the focus was placed on the mediation analysis in the present study, the amplitude changes of different components could certainly contribute to the patterns seen in the mediation analysis. This could be addressed by a 3 (conditions) x 3 (intensity levels) repeated measures ANOVA with the LEP components of interest as dependent measures.

3. The topographic maps of N1 component in Figure 3 and in Supplementary figure do not show the typical bilateral temporal negative maxima which have been shown in a number of previous EEG studies. The N1 component is often difficult to assess using scalp data due to being overridden by a much stronger N2 component which already picks up when the N1 component reaches its maximum. While Authors correctly evaluated the N1 component using a bipolar lead C4-Fz (maybe, the recommended T8-Fz lead could have been used?), the map of N1 component in Figure 3 (N1) shows actually the topography of the N2 component. Would a map of N1 component with data referenced to Fz show a topographic map consistent with the N1 component? Could the ICA applied during the pre-processing stage have removed the comparatively small negative components in bilateral temporal regions of the scalp?

4. The interpretation of data is limited by the absence of a source reconstruction of LEP data. Interpretations on to the possible generators refer to selected findings from previous studies and since there is a variability in locations of sources of different LEP components among published studies, it is not clear which previous source localisation studies should apply to the present data. For instance, Author's suggestion that the N1 component refers to the activation of operculo-insular cortex has limited support in the present study given the absence of a negative potential maximum in the temporal electrodes.

Reviewer #2 (Remarks to the Author):

Tiemann and colleagues report the results of a study that used mediation analysis to identify the EEG-based mediators of pain reports, reaction time, and skin conductance responses to painful laser stimulation. The paper is very clearly written, analyses are appropriate and sound, and this is

likely to be of interest to the pain community as it builds on our current understanding of brain mediators of pain and pain-related responses. That said, there are several major issues that should be addressed before the manuscript is suitable for publication, including potential confounds.

1) The task included separate blocks in which subjects were asked to either rate pain, perform a motor task, or passively receive pain. The authors therefore cannot conclude anything about independent versus serial processes, as the cognitive demands of the tasks varied between blocks, and therefore the underlying neural systems can be assumed to vary as well. There is no way whatsoever to conclude time precedence or independence of these signatures when separately measured if the underlying cognitive states varied. This is a fundamental flaw in the paper, but I was surprised to see that in the methods section, the authors state that “a fourth condition with a combination of the perceptual and motor tasks was performed, which was not included in this present analysis” (p.15). The only way these conclusions of serial independent pathways for pain, motor, and autonomic responses should be accepted as such is if similar separations are seen in this condition. It should be simple to measure reaction time, pain, and SCR in a task where a subject depresses the button and then reports pain, while SCR is measured. Given the fact that this condition (or at least one where motor and pain were both recorded) I suspect the authors already have these data and have run these analyses but conclusions varied from the mediations based on the separate conditions. If this is the case (i.e. that distinct EEG features mediate pain, autonomic, and motor responses only when cognitive demands vary), then we can conclude nothing about their serial independence, and they’re likely to reflect task demands instead. If however these conclusions replicate when measured simultaneously, then this would be a very important contribution to the literature. This analysis must be included in order to interpret these findings and make statements concluding that these dimensions are “partially independent rather than serial processes.”

2) In a related point, it is unknown whether the outcome measures are actually dissociable, since they are measured at separate times. It might be that the brain processes that predict pain and SCR are indistinguishable, but pain was only measured when subjects were told to attend to the potential rating, while SCR was only measured in a passive context. Again, the concern is that the findings are unique to the cognitive task. The authors should report the correlation structure in the context in which all outcomes are measured simultaneously in order to determine whether these responses are truly distinct.

3) The introduction and discussion read as though this is the first paper to use mediation analysis to measure the links between noxious input and pain or pain-related responses. The authors acknowledge the work of Atlas et al. (Pain, 2014) and Woo et al. (PLoS Biol 2015, Nature Communications 2017) but only as examples of the fact that mediation has “increasingly been applied to neuroimaging data” (p.7 and p.11, identical statements). They ignore the fact that these studies were directly designed to test which brain processes link noxious input and subjective pain, similar to the present study. The current paper builds on this work by a) featuring distinct outcomes that might be dissociable from pain (motor responses and SCR; but see point 2 above), b) using brief stimuli that better allow for separation in time and therefore stronger assertions of directionality in the mediation framework, and c) using EEG rather than fMRI, which provides better temporal resolution and can distinguish contributors to the LEP as well as oscillatory responses. Rather than dismissing this work in an effort to seem more novel, the authors should acknowledge these studies in the introduction and compare results, for instance in the brain systems that have been previously linked to N1, N2, and P2 waves (p. 11).

4) The title is somewhat misleading in light of a large body of work now focusing on pattern analyses using machine learning approaches. There are no “patterns” in this paper. A different term should be used.

5) There are no path a effects on N1 in any of the conditions (Table 2). The authors should be commended on their discussion of covariance in multilevel mediation, but still need to reconcile their failure to see any effects of intensity on N1 given its central role in pain and nociception and its putative importance in predicting motor and autonomic responses here.

Reviewer #3 (Remarks to the Author):

The manuscript "Distinct patterns of brain activity mediate the perceptual, motor and autonomic dimensions of pain" by Tiemann et al. describes a scalp EEG study investigating perceptual, motor, and autonomic responses to brief thermocceptive stimuli in healthy human participants. By using multilevel mediation analysis, the authors aimed to determine how different features of the EEG responses contribute to different dimensions of pain. The main conclusion they draw from their analysis is that the perceptual, motor, and autonomic dimensions of pain are not processed serially, and are partially independent.

Despite addressing a clear and original question, the study has a very important limitation, that is, the lack of a non-painful control stimulus. In other words, the study design does not allow to conclude whether the obtained perceptual, motor, and autonomic responses are actually related to pain. In fact, it is quite likely that similar independent responses would be obtained using non-painful/non nociceptive yet highly salient and/or arousing stimuli (e.g. loud sounds, bright flashes, etc.). These stimuli may also signal a threat, despite the fact that they are not painful. Indeed, reactions to such stimuli necessarily have perceptual, motor, and autonomic components, regardless of their being or not painful.

Crucially, the absence of any sort of control stimulus does not allow us to conclude whether the obtained responses are a) actually pain-related; b) related to nociception/spinothalamic activation but not necessarily to pain; c) related to the processing of thermal information/heat (which is not necessarily painful); d) related to the salience of the stimuli. Because these different aspects cannot be disentangled in this study, the observation that "Distinct patterns of brain activity mediate the perceptual, motor and autonomic dimensions of pain" is an overstatement. The title should rather read: "Distinct patterns of brain activity mediate perceptual, motor, and autonomic responses to thermal nociceptive stimuli" - a conclusion that is probably less enticing.

Further comments:

- P. 3: The authors state "It has further been shown that gamma oscillations provide complementary information which can be particularly closely related to the perception of pain". Although it is true that several studies have shown a relationship between the perception of pain and gamma oscillations, these findings should be interpreted with caution. First, because gamma oscillations can also be observed in response to non-painful stimuli (e.g., Fardo et al. 2017, *Journal of Neurophysiology*). Second, because at least in some circumstances, the amplitude of pain-related gamma oscillations can be dissociated from pain perception (e.g., Liberati et al. 2018, *Scientific Reports*).

- P. 18: How were the time window and the frequency range of gamma oscillations chosen? In particular, the 70-90 Hz frequency range seems very restricted, considering that any activity >40 Hz is generally regarded as gamma, and oscillations at frequencies higher than 90 Hz can normally be observed in response to sensory stimuli (e.g., see Zhang et al. 2012).

- Gamma oscillations are quite difficult to observe on the scalp EEG, and a large number of subjects do not show a clear gamma response to thermocceptive stimuli. In Fig. 3 only an average gamma response is shown. It would be important to understand how many participants out of 51 clearly showed this response, and how many were "non-responders". In particular, it would be important that the authors shared the data of the individual participants which have led to these results, e.g. on a public repository.

- P. 18: It is not clear why theta oscillations were not analyzed, and what the authors mean with "Responses at theta frequencies were not analyzed as they are known to mainly represent pain-evoked activity". Aren't other oscillations (e.g. gamma) also considered by the authors as pain-

evoked activity?

- P. 12: "The relationship between perceptual, motor and autonomic dimensions of pain is also relevant for the understanding of chronic pain". This statement is reasonable, but represents a large "leap" in the context of this study, in which only very brief noxious stimuli (1 ms) are used. These brief stimuli are predominantly mediated by quickly-adapting thermoreceptors. In contrast, the development of chronic pain has been related to slowly-adapting C fiber thermoreceptors (e.g., see Wooten et al. 2014, Nature Communications). It is therefore likely that the processing of transient nociceptive stimuli and the development of chronic pain underlie very different processes.

- All stimuli were applied on the left hand, and motor responses were given using the right hand. How many participants were right/left handed? Wouldn't this aspect interfere with the reaction times?

- The authors mention that the EEG was acquired while participants kept their eyes closed. This is likely to cause an alteration in the ongoing EEG oscillations, which should be taken into account (e.g. see Barry et al. 2007, Clinical Neurophysiology).

- P. 16: "Trials in which no or a delayed motor response occurred (reaction times > 650 ms) were discarded". Can the authors explain the reason for excluding these trials? Was it to avoid considering sensations mediated by C fibers?

- P. 17: How many trials were rejected due to artifact contamination?

- Fig. 1 would be clearer with a timeline showing the duration of the stimuli and the variable ISI.

- Fig. 2 would be more informative if the individual data were shown in a scatterplot. Bar graphs only showing average values are deceiving - especially considering that the standard deviations appear to be quite large.

- The authors state that "The study was approved by the local ethics committee and conducted in conformity with the Declaration of Helsinki" (p. 14). According to the Declaration of Helsinki, "Every research study involving human subjects must be registered in a publicly accessible database before recruitment of the first subject". Can the Authors confirm that this is the case?

Response to referees

Reviewer #1

The submitted study analysed the associations between individual components of LEPs (N1, N2, P2 and gamma band response) in three tasks designed to capture pain intensity, reaction time and autonomic responses. Laser stimuli of three different intensities were used to elicit responses of monotonically increasing amplitudes, and a complex mediation analysis was employed to attribute subsets of LEP responses to either motor, autonomic or subjective responses. Results show that the motor responses were mediated primarily by the gamma responses, subjective responses by the N2 and P2 LEP components, and autonomic responses by the N1 LEP component. Results are novel in teasing apart different types of responses to noxious stimuli and their cortical mediators. The study is well-conducted and important as it sheds light on the functional significance of LEP components. However, there are few points needing amendments and/or explanations which limits reviewer's possibility to comment on a wider impact of the study.

We thank the reviewer for kindly acknowledging that our study is well-conducted and important. In the following, we will provide amendments and explanations to further clarify the impact of the study.

1. The three experiments have been conducted in blocks each requiring the participants to carry out a different task. Therefore, factors such as motor readiness or focusing attention towards the source of pain (rather than on e.g., motor response) might have affected the mediation weights. For instance, the N1 component in LEPs occurring during periods of a high motor response expectancy shows an increased amplitude compared to periods with low level of motor response expectancy (Stancak et al., Behav. Brain Res., 2012). Similarly, focusing attention on the hand receiving a noxious stimulus, a condition perhaps similar to the "autonomic" condition in the present study, has been shown to reduce the N2/P2 components (Longo et al., J. Neurosci., 2009). Could the differences in mediation weights among the three conditions be attributed to the fact that they related to different tasks and instruction sets?

We appreciate this important comment. The perceptual, motor and autonomic responses to noxious stimuli were measured in separate conditions with different tasks and instructions. We therefore agree that, based on these conditions, it is not possible to ultimately rule out that differences in tasks and instructions have contributed to the observed differences in mediation patterns between conditions. We are, however, confident that the present findings are more likely to represent basic differences in how the brain translates noxious stimuli into perceptual, motor and autonomic responses than task effects.

First, as exemplified by the studies cited by the reviewer, task effects on brain responses to noxious stimuli have been shown to influence the amplitudes of brain responses. In the present analysis, task effects on amplitudes were controlled by z-transformation of brain responses. Thus, an influence of task instructions on mediation weights would have to be independent from an influence on amplitudes. Second, to fully explain the present results, tasks and task instructions would have to fundamentally change the processing steps and processing hierarchy which underlies the translation of a noxious stimulus into

a certain response. In our view, such a task-induced fundamental change of the processing hierarchy of a sensory stimulus is unlikely.

Nevertheless, the reviewer's comments are well-founded. Moreover, even the *combined* condition does not allow for the unequivocal differentiation of task effects and mediation effects underlying the translation of a noxious stimulus into different responses (see our reply to the first comment of reviewer 2 for a discussion of the *combined* condition). We can therefore not rule out that tasks and instructions have influenced mediation patterns. We have now explicitly addressed this important limitation in the discussion section (page 14, last paragraph). The paragraph reads as follows.

"Some limitations apply to the present findings and their interpretation. First, an influence of task effects on mediation patterns serving the translation of noxious stimuli into perceptual, motor and autonomic responses cannot be ruled out. At first glance, the lack of mediation effects in the *combined* condition suggests such task effects. However, in this condition, influences of motor responses on perceptual (May et al., Sci Rep, 2017) and autonomic responses (Boucsein, 2012, "Electrodermal Activity", Springer 2nd edition chapter 2.2.5.2, page 141) are likely to prevent the separate assessment of the underlying processes. Moreover, task effects on brain responses to noxious stimuli have been shown to manifest as amplitude differences (e.g. Stancak et al., Behav Brain Res, 2012; Nakata et al., Neuroimage, 2009; Longo et al., J Neurosci, 2009; Legrain et al., Pain, 2002). In the present study, amplitude differences between conditions were adjusted prior to the analysis. The only way different tasks could influence mediation weights would therefore be an influence on mediation weights independent from an influence on amplitudes. Moreover, it appears unlikely that different tasks not only modulate, but fundamentally change the processing steps and processing hierarchy which translates a noxious stimulus into a certain response."

2. Along the same vein as (1), did LEPs differ between the three conditions? While the focus was placed on the mediation analysis in the present study, the amplitude changes of different components could certainly contribute to the patterns seen in the mediation analysis. This could be addressed by a 3 (conditions) x 3 (intensity levels) repeated measures ANOVA with the LEP components of interest as dependent measures.

Following the reviewer's suggestion, we calculated a 3 (intensity levels) x 3 (conditions) repeated measures ANOVA for each brain response. The results indicate that noxious stimulus intensity significantly influenced the amplitudes of all four brain responses (N1: $F_{(2,98)} = 6.72$, $p = 0.002$; N2: $F_{(2,76)} = 3.83$, $p = 0.04$; P2: $F_{(2,98)} = 27.96$, $p < 0.001$; gamma: $F_{(2,98)} = 11.56$, $p < 0.001$; Greenhouse-Geisser corrected where necessary). Moreover, condition significantly influenced amplitudes of N1 ($F_{(2,98)} = 14.78$, $p < 0.001$) and gamma ($F_{(2,98)} = 6.75$, $p = 0.002$) but not of N2 ($F_{(2,98)} = 1.87$, $p = 0.16$) and P2 ($F_{(2,98)} = 0.12$, $p = 0.89$) responses. No significant interaction of stimulus intensity and condition was observed ($p > 0.06$ for all responses). Thus, amplitudes of brain responses were influenced by stimulus intensity and in part by condition.

However, prior to mediation analysis, all response amplitudes were z-transformed across participants and trials for each condition. This procedure controls for amplitude differences between conditions so that the results of the mediation analysis cannot be influenced by amplitude differences. Figure 1 illustrates the results of this procedure by

showing the amplitudes after z-transformation which were entered into the mediation analysis. We have added the results of the ANOVA to the manuscript (p. 6, last paragraph) and clarified the important point that amplitude differences between conditions cannot explain differences in mediation effects between conditions on p. 14, last paragraph.

Figure 1. Amplitudes of brain responses (z-standardized, baseline corrected (absolute baseline)) as entered into the mediation analysis.

3. The topographic maps of N1 component in Figure 3 and in Supplementary figure do not show the typical bilateral temporal negative maxima which have been shown in a number of previous EEG studies. The N1 component is often difficult to assess using scalp data due to being overridden by a much stronger N2 component which already picks up when the N1 component reaches its maximum. While Authors correctly evaluated the N1 component using a bipolar lead C4-Fz (maybe, the recommended T8-Fz lead could have been used?), the map of N1 component in Figure 3 (N1) shows actually the topography of the N2 component. Would a map of N1 component with data referenced to Fz show a topographic map consistent with the N1 component? Could the ICA applied during the pre-processing stage have removed the comparatively small negative components in bilateral temporal regions of the scalp?

As suggested by the reviewer, we performed additional analyses to check whether the choice of another EEG electrode and/or the omission of artifact correction using ICA might yield a more discernible N1 response.

First, we calculated waveforms of brain responses with a temporal-frontal electrode montage (T8-Fz) and compared them to the C4-Fz montage used in the present study (Figure 2). The comparison shows that the N1 waveforms using the C4-Fz montage shows a clearer peak in the predefined N1-time window than the T8-Fz montage. Moreover, in the C4-Fz montage, the N1 response can be better differentiated from the subsequent N2 and P2 waves than in the T8-Fz montage. This comparison confirms that the N1 response can be better detected using a central-frontal montage (C4-Fz) which is in accordance with previous findings and recommendations (Hu et al., Neuroimage, 2010).

Figure 2. N1 waveforms in the perceptual, motor, and autonomic condition calculated based on central-frontal (upper row) or temporal-frontal (lower row) montage. Marked time periods indicate periods chosen to quantify N1 responses (150-180 ms).

Second, to check whether the ICA has diminished or even removed the N1 component, we calculated N1 topographies without preceding ICA and compared them to N1 topographies with preceding ICA (Figure 3). The comparison shows that the resulting N1 topographies closely resemble each other. Moreover, the ICA results in lower amplitudes and less frontal artefacts related to eye movements. The lateralization of the central negativity, however, is not significantly affected by applying ICA to the data.

Figure 3. Topographies of the N1 response (150-180 ms, C4-Fz) in the perceptual, motor, and autonomic condition calculated on data with (upper row) or without (lower row) preceding ICA.

These analyses show that the choice of the EEG electrode and/or the ICA has not diminished the N1 response. Moreover, the waveforms presented in the manuscript show a clearly defined peak, the latency of which coincides with the predefined and N1-typical (e.g., Hu et al., Neuroimage, 2010) time-window of 150 - 180 ms. In terms of latency and amplitude, this peak can be clearly differentiated from later activity. Finally, only averaged activity from the predefined time-window from 150 - 180 ms was entered into subsequent mediation analyses. This approach ensured that only activity centered around the N1 peak and no activity later than 180 ms was included in the analysis. We are therefore confident that the present approach is a suitable approach to analyze the N1 response.

4. The interpretation of data is limited by the absence of a source reconstruction of LEP data. Interpretations on to the possible generators refer to selected findings from previous studies and since there is a variability in locations of sources of different LEP components among published studies, it is not clear which previous source localisation studies should apply to the present data. For instance, Author's suggestion that the N1 component refers to the activation of operculo-insular cortex has limited support in the present study given the absence of a negative potential maximum in the temporal electrodes.

We agree that source reconstruction of EEG data can yield valuable but inherently ambiguous information. With respect to LEP, different source localization approaches have been used ranging from dipole models with a few equivalent current dipoles to distributed source models. However, no standard source reconstruction approach of LEP has been established so far. We have therefore deliberately refrained from source reconstruction and performed all analyses in electrode space. Only when considering the most likely generators of the different brain responses in the discussion, we refer to source reconstruction studies. In particular, we refer to a review (Garcia-Larrea et al., Neurophysiol Clin, 2003) and a recent study using intracerebral recordings in epileptic patients (Bradley et al., Hum Brain Mapp, 2017) which provides convincing evidence on the generators of LEP. Nevertheless, even these data are not unambiguous and source reconstruction might also change with stimulation parameters and task demands. We have therefore now explicitly mentioned the important limitation of source reconstruction when reviewing the above-mentioned studies in the discussion (p. 12, first paragraph).

Reviewer #2

Tiemann and colleagues report the results of a study that used mediation analysis to identify the EEG-based mediators of pain reports, reaction time, and skin conductance responses to painful laser stimulation. The paper is very clearly written, analyses are appropriate and sound, and this is likely to be of interest to the pain community as it builds on our current understanding of brain mediators of pain and pain-related responses. That said, there are several major issues that should be addressed before the manuscript is suitable for publication, including potential confounds.

We thank the reviewer for this positive overall assessment of the manuscript. We are confident that the revised version appropriately addresses the major issues raised by the reviewer.

1) The task included separate blocks in which subjects were asked to either rate pain, perform a motor task, or passively receive pain. The authors therefore cannot conclude anything about independent versus serial processes, as the cognitive demands of the tasks varied between blocks, and therefore the underlying neural systems can be assumed to vary as well. There is no way whatsoever to conclude time precedence or independence of these signatures when separately measured if the underlying cognitive states varied. This is a fundamental flaw in the paper, but I was surprised to see that in the methods section, the authors state that “a fourth condition with a combination of the perceptual and motor tasks was performed, which was not included in this present analysis” (p.15). The only way these conclusions of serial independent pathways for pain, motor, and autonomic responses should be accepted as such is if similar separations are seen in this condition. It should be simple to measure reaction time, pain, and SCR in a task where a subject depresses the button and then reports pain, while SCR is measured. Given the fact that this condition (or at least one where motor and pain were both recorded) I suspect the authors already have these data and have run these analyses but conclusions varied from the mediations based on the separate conditions. If this is the case (i.e. that distinct EEG features mediate pain, autonomic, and motor responses only when cognitive demands vary), then we can conclude nothing about their serial independence, and they’re likely to reflect task demands instead. If however these conclusions replicate when measured simultaneously, then this would be a very important contribution to the literature. This analysis must be included in order to interpret these findings and make statements concluding that these dimensions are “partially independent rather than serial processes.”

We are very grateful for this important comment. We fully agree that, based on the *perception*, *motor* and *autonomic* conditions, task and instruction effects on mediation patterns cannot be ruled out.

In order to differentiate task-related effects from differences in the mediation effects translating a noxious stimulus into perceptual, motor and autonomic responses, we had originally included a fourth *combined* condition in our experiment. In this condition, skin conductance was recorded and participants were asked to react to each stimulus and to subsequently rate it. However, during performance and preliminary analysis of the experiment, we realized that it is difficult to unequivocally interpret the *combined* condition. In particular, it has been shown that motor responses influence pain ratings (May et al., Sci Rep, 2017) as well as skin conductance responses (Boucsein, 2012, “Electrodermal Activity”, Springer 2nd edition chapter 2.2.5.2, page 141). These

influences likely interfere with a clear differentiation of mediation patterns underlying the translation of noxious stimuli into perceptual, motor and autonomic responses in the *combined* condition.

However, we understand that it is important to know the results of the *combined* condition which we have analyzed in the meantime as suggested by the reviewer (Table 1). The results of the mediation analyses showed that the pattern of significant results for path a, b, c and c' effects in the *combined* condition were basically identical to the *perception*, *motor* and *autonomic* conditions. However, in the *combined* condition, no significant mediation effects were found for any brain response or pain dimension. Further analysis revealed that this is likely due to higher variability of path a*b coefficients for the *combined* condition. Indeed, standard errors for path a*b coefficients were significantly higher for the *combined* compared to single conditions (two-tailed paired t-test, $t_{(11)} = -5.4$, $p < 0.001$). Moreover, the overall pattern of path a*b coefficients was not significantly different between *combined* and single conditions (two-tailed paired t-test, $t_{(11)} = 0.07$, $p = 0.95$), but instead significantly correlated (Pearson correlation, $r = 0.66$, $p = 0.02$). We have included the results of the *combined* condition in the manuscript (page 8, third paragraph, Supplementary Table 1).

		Perception				Motor				Autonomic			
		β	SE	Z	p	β	SE	Z	p	β	SE	Z	p
N1	a	.0068	.0178	.3541	.7233	.0068	.0176	.3939	.6936	.0068	.0177	.4017	.6879
	b	-.0920	.0188	-3.295	.0013	.1569	.0186	3.921	.0004	-.0421	.0111	-3.459	.0005
	c'	.1647	.0159	3.907	.0001	-.0833	.0196	-3.538	.0005	.0957	.0103	3.723	.0002
	c	.1636	.0162	3.982	.0001	-.0843	.0190	-3.580	.0005	.0977	.0103	3.728	.0002
	a*b	-.0003	.0012	-3.430	.9755	-.0011	.0021	-5.200	.6031	.0007	.0007	1.144	.2528
N2	a	.0108	.0218	.5069	.7233	.0108	.0216	.4784	.6936	.0108	.0218	.4720	.6879
	b	-.1101	.0226	-3.485	.0010	.0929	.0297	2.677	.0149	-.0943	.0180	-3.467	.0005
	c'	.1620	.0155	4.098	.0001	-.0784	.0183	-3.668	.0005	.1007	.0103	3.672	.0002
	c	.1636	.0164	4.066	.0001	-.0843	.0189	-3.539	.0005	.0977	.0103	3.694	.0002
	a*b	.0000	.0017	-.0177	.9859	-.0018	.0023	-8.070	.5596	-.0023	.0014	-1.739	.1640
P2	a	.0586	.0199	3.105	.0038	.0586	.0201	3.157	.0032	.0586	.0199	3.162	.0031
	b	.0664	.0231	2.954	.0031	-.0811	.0342	-2.444	.0194	.0628	.0186	3.701	.0004
	c'	.1531	.0150	3.846	.0001	-.0747	.0184	-3.502	.0005	.0913	.0104	3.750	.0002
	c	.1636	.0164	4.089	.0001	-.0843	.0190	-3.518	.0005	.0977	.0104	3.732	.0002
	a*b	.0027	.0016	1.717	.1719	-.0035	.0026	-1.335	.5008	.0020	.0014	1.449	.1966
gamma	a	.0268	.0081	3.717	.0008	.0268	.0081	3.725	.0008	.0268	.0081	3.685	.0009
	b	.1179	.0321	3.513	.0010	-.0631	.0514	-1.203	.2289	.1580	.0329	4.016	.0002
	c'	.1569	.0157	3.839	.0001	-.0788	.0177	-3.648	.0005	.0906	.0103	3.775	.0002
	c	.1636	.0165	4.010	.0001	-.0843	.0189	-3.589	.0005	.0977	.0103	3.722	.0002
	a*b	.0020	.0011	1.891	.1719	-.0016	.0015	-1.149	.5008	.0022	.0011	2.188	.1146

Table 1. Second-level statistics for the mediation analyses of N1, N2, P2 and gamma responses in the *combined* condition. All p-values are FDR-corrected. Significant effects of paths a, b, c and c' are marked in bold. Significant mediation effects (path a*b) are marked in red. β , regression coefficient; SE, standard error.

These new findings suggest an overall similar pattern of results of the mediation analysis in the *combined* and single conditions but an increased variability of mediation effects in the *combined* condition. This increase in variability might be due to the inextricable relationships between perceptual, motor and autonomic responses which are exemplified by an influence of motor responses on perceptual (May et al., Sci Rep, 2017) and autonomic (Boucsein, 2012, "Electrodermal Activity", Springer 2nd edition chapter 2.2.5.2, page 141) responses. Variability in mediation effects might be further increased by higher task demands in the *combined* as compared to the single conditions. Such a difference is suggested by significantly longer reaction times in the *combined* than in the *motor* condition (*combined*, 412 ± 55 ms; *motor*, 363 ± 50 ms; two-tailed paired t-test: $t(45) = 6.79$, $p < 0.001$). Taken together, the pattern of significant correlation of beta coefficients but significantly increased standard errors for the *combined* versus single conditions seems to make the increased variability the most parsimonious explanation for the lack of significant mediation effects in the *combined* condition.

Based on these findings and the following considerations, we are confident that the present findings more likely represent basic differences in how the brain translates noxious stimuli into perceptual, motor and autonomic responses than task effects. First, although the *combined* condition does not allow for proving or disproving task effects on the observed mediation patterns, the findings are, in principle, compatible with the results of the single conditions. Second, task effects such as motor preparation (e.g., Nakata et al., Neuroimage, 2009; Stancak et al., Behav Brain Res, 2012) or attention (e.g., Legrain et al., Pain, 2002; Longo et al., J Neurosci, 2009) on brain responses to noxious stimuli have been shown to manifest as amplitude modulations. In the present analysis, task effects on amplitudes were controlled by z-transformation of brain responses. Thus, an influence of task instructions on mediation weights would have to be independent from an influence on amplitudes. Third, to fully explain the present results, task instructions would have to fundamentally change the processing steps and processing hierarchy which underlies the translation of a noxious stimulus into a certain response. In our view, such a task-induced fundamental change of the processing hierarchy of a sensory stimulus is unlikely in our view.

We, however, agree that task effects on mediation patterns cannot be ultimately ruled out. We have therefore toned down our statements regarding the independence and serial organization throughout the manuscript. Moreover, we have added a paragraph to the discussion which addresses this important limitation (page 14, last paragraph). The paragraph reads as follows.

"Some limitations apply to the present findings and their interpretation. First, an influence of task effects on mediation patterns serving the translation of noxious stimuli into perceptual, motor and autonomic responses cannot be ruled out. At first glance, the lack of mediation effects in the *combined* condition suggests such task effects. However, in this condition, influences of motor responses on perceptual (May et al., Sci Rep, 2017) and autonomic responses (Boucsein, 2012, "Electrodermal Activity", Springer 2nd edition chapter 2.2.5.2, page 141) are likely to prevent the separate assessment of the underlying processes. Moreover, task effects on brain responses to noxious stimuli have been shown to manifest as amplitude differences (e.g. Stancak et al., Behav Brain Res, 2012; Nakata et al., Neuroimage, 2009; Longo et al., J Neurosci, 2009; Legrain et al., Pain, 2002). In the present study, amplitude differences between conditions were

adjusted prior to the analysis. The only way different tasks could influence mediation weights would therefore be an influence on mediation weights independent from an influence on amplitudes. Moreover, it appears unlikely that different tasks not only modulate, but fundamentally change the processing steps and processing hierarchy which translates a noxious stimulus into a certain response.”

2) In a related point, it is unknown whether the outcome measures are actually dissociable, since they are measured at separate times. It might be that the brain processes that predict pain and SCR are indistinguishable, but pain was only measured when subjects were told to attend to the potential rating, while SCR was only measured in a passive context. Again, the concern is that the findings are unique to the cognitive task. The authors should report the correlation structure in the context in which all outcomes are measured simultaneously in order to determine whether these responses are truly distinct.

As suggested, we have assessed the correlation structure of the three outcomes (pain ratings, reaction times, SCR) in the combined condition. To do so, we correlated single trial outcome measures for each subject yielding three r values per subject. Subsequently, we averaged the correlation coefficients across participants. The results show weak correlations between pain ratings and reaction times ($r = -0.24$), reaction times and SCR ($r = -0.04$), and pain ratings and SCR ($r = 0.27$) corresponding to small effect sizes (Cohen, 1977, “Statistical power analysis for behavioral sciences”, Hillsdale, NJ). Moreover, we calculated the coefficient of determination (r^2), in order to evaluate which proportion of the variance in one outcome measure can be predicted by the respective other outcome measures. R^2 was 9% + 10 of shared variance between pain ratings and reaction times, 4% + 4 of shared variance between reaction times and SCRs, and 13% + 11 of shared variance between pain ratings and SCRs. Thus, we deem the three outcome measures to be sufficiently distinct to be statistically discernible. We have included a new Supplementary Figure 1 showing pain ratings, reaction times and SCR of the combined condition and added the above-mentioned results of the correlation analyses to the figure legend.

3) The introduction and discussion read as though this is the first paper to use mediation analysis to measure the links between noxious input and pain or pain-related responses. The authors acknowledge the work of Atlas et al. (Pain, 2014) and Woo et al. (PLoS Biol 2015, Nature Communications 2017) but only as examples of the fact that mediation has “increasingly been applied to neuroimaging data” (p.7 and p.11, identical statements). They ignore the fact that these studies were directly designed to test which brain processes link noxious input and subjective pain, similar to the present study. The current paper builds on this work by a) featuring distinct outcomes that might be dissociable from pain (motor responses and SCR; but see point 2 above), b) using brief stimuli that better allow for separation in time and therefore stronger assertions of directionality in the mediation framework, and c) using EEG rather than fMRI, which provides better temporal resolution and can distinguish contributors to the LEP as well as oscillatory responses. Rather than dismissing this work in an effort to seem more novel, the authors should acknowledge these studies in the introduction and compare results, for instance in the brain systems that have been previously linked to N1, N2, and P2 waves (p. 11).

We are very sorry that the introduction created the impression that we wanted to dismiss previous fMRI studies which was not at all our intention. We fully agree that the previous

fMRI studies are an important basis of the present study and that it is appropriate to introduce these studies in more detail. Moreover, it is helpful to clarify how the present study complements and extends these studies. We are therefore very thankful for the reviewer's constructive suggestions on how the present study builds upon and extends the findings of previous studies. Based on these suggestions we have therefore substantially revised the introduction (p.3, last paragraph; p.4, first paragraph; p.4, last paragraph). We are convinced that the introduction has thereby improved significantly and gives now a more balanced summary of previous work. Moreover, we have compared the present results to previous findings of fMRI studies in the discussion (p. 12, last paragraph).

4) The title is somewhat misleading in light of a large body of work now focusing on pattern analyses using machine learning approaches. There are no "patterns" in this paper. A different term should be used.

We understand that the term "pattern" is now often used with respect to machine learning approaches and have therefore carefully considered alternative terms. However, we haven't found an equally appropriate term and still think that the term "pattern" is most appropriate as our results show not only the relationships of single brain responses to different dimensions of pain, but rather how patterns of different brain responses relate to different dimensions of pain. These patterns are summarized and illustrated in Figure 6 of the manuscript. We would therefore prefer to retain the term "pattern" in the title but are open to specific suggestions for avoiding the misunderstanding that the present study includes machine learning approaches.

5) There are no path a effects on N1 in any of the conditions (Table 2). The authors should be commended on their discussion of covariance in multilevel mediation, but still need to reconcile their failure to see any effects of intensity on N1 given its central role in pain and nociception and its putative importance in predicting motor and autonomic responses here.

We thank the reviewer for acknowledging the discussion of covariance in multilevel mediation in the manuscript and agree that the lack of path a effects on the N1 response is unexpected. To further test for an effect of stimulus intensity on the N1 response, we calculated a 3 (intensity levels) x 3 (conditions) repeated measures ANOVA with N1 amplitude as dependent variable. The results show a significant main effect of intensity ($F_{(2, 98)} = 6.72, p = .002$) as well as of condition ($F_{(2, 98)} = 14.78, p < .001$). No significant interaction effect between intensity and condition was found ($F_{(4, 196)} = 0.06, p = .99$). These findings indicate that the N1 response is significantly influenced by stimulus intensity. However, this effect is not statistically significant when the conditions are analyzed separately (as in the mediation analyses) but only when they are analyzed together (as in the ANOVA). We have added the results of the ANOVA on p. 6, last paragraph.

Reviewer #3

The manuscript "Distinct patterns of brain activity mediate the perceptual, motor and autonomic dimensions of pain" by Tiemann et al. describes a scalp EEG study investigating perceptual, motor, and autonomic responses to brief thermocceptive stimuli in healthy human participants. By using multilevel mediation analysis, the authors aimed to determine how different features of the EEG responses contribute to different dimensions of pain. The main conclusion they draw from their analysis is that the perceptual, motor, and autonomic dimensions of pain are not processed serially, and are partially independent.

Despite addressing a clear and original question, the study has a very important limitation, that is, the lack of a non-painful control stimulus. In other words, the study design does not allow to conclude whether the obtained perceptual, motor, and autonomic responses are actually related to pain. In fact, it is quite likely that similar independent responses would be obtained using non-painful/non nociceptive yet highly salient and/or arousing stimuli (e.g. loud sounds, bright flashes, etc.). These stimuli may also signal a threat, despite the fact that they are not painful. Indeed, reactions to such stimuli necessarily have perceptual, motor, and autonomic components, regardless of their being or not painful.

Crucially, the absence of any sort of control stimulus does not allow us to conclude whether the obtained responses are a) actually pain-related; b) related to nociception/spinothalamic activation but not necessarily to pain; c) related to the processing of thermal information/heat (which is not necessarily painful); d) related to the salience of the stimuli. Because these different aspects cannot be disentangled in this study, the observation that "Distinct patterns of brain activity mediate the perceptual, motor and autonomic dimensions of pain" is an overstatement. The title should rather read: "Distinct patterns of brain activity mediate perceptual, motor, and autonomic responses to thermal nociceptive stimuli" - a conclusion that is probably less enticing.

We thank the reviewer for the overall positive assessment of our manuscript. We agree that the specificity of neural processes underlying pain is a highly interesting and relevant topic which has been debated in pain research for a long time (reviewed in Moayedi and Davis, *J Neurophysiol*, 2013). During recent years, seminal EEG (Mouraux and Iannetti, *J Neurophysiol*, 2009) and fMRI (Mouraux et al., *Neuroimage*, 2011) studies have shown that most brain responses to noxious stimuli are not specific for pain but rather reflect a brain system involved in detecting, orienting attention towards, and reacting to salient sensory events (Legrain et al., *Prog Neurobiol*, 2011). So far, no spatially, temporally, or spectrally defined feature of brain activity which is specific for pain has been demonstrated. Hence, it is questionable whether any pain specific feature of brain activity exists. Correspondingly, studies claiming pain specificity (e.g., Segerdahl et al., *Nat Neurosci*, 2015) have been heavily criticized (e.g., Davis et al., *F1000Res*, 2015).

Furthermore, we concur with the reviewer that the present study does not show pain specificity. However, throughout the manuscript, we do not claim pain specificity and pain specificity is not the central point of the study. When nevertheless considering control experiments addressing specificity for pain, nociception, thermoception and salience, several requirements must be fulfilled. First, auditory, visual, tactile, thermal and nociceptive control stimuli are needed to prove modality specificity. Second, all control

stimuli must not be painful but equally salient. Third, all control stimuli have to have a similar duration to allow for the comparison of the temporal patterns of brain responses. Fourth, the intensities of all control stimuli need to be similarly graded to allow for comparing mediation analysis findings. Fifth, the spatial-temporal-spectral patterns of brain responses to the different control stimuli need to be, in principle, similar including N1, N2, P2 and gamma responses. Otherwise, the comparison of the patterns of brain activity translating the stimuli into perceptual, motor and autonomic responses is confounded by differences in the patterns of brain responses across modalities.

Considering these requirements, several obstacles need to be overcome. First, in the present context, the intensity, duration, and salience of auditory, visual, tactile, thermal and nociceptive stimuli cannot be properly matched to the painful stimuli. For instance, we cannot imagine a thermal stimulus of 1 ms duration which is graded across three intensities and non-painful but equally salient to the painful stimulus. Second, the spatial-spectral-temporal patterns of brain responses differ across modalities which hampers a proper comparison of mediation patterns for perceptual, motor and autonomic responses across modalities. For instance, in human EEG recordings, visual (Murty et al., *J Neurosci*, 2018) and auditory (Polomac et al., *Brain Topogr*, 2015) stimuli yield gamma responses at other locations and latencies than the painful stimuli. Thus, a direct comparison of mediation patterns across these modalities is not possible. Third, specificity of brain function is an inherently difficult and controversial issue. Similarities and differences in brain activity related to different stimuli and tasks do not necessarily prove or disprove specificity as, in principle, an infinite number of stimuli and tasks is possible. For example, in the present context, when controlling for salience, specificity with regard to other factors such as unpleasantness, arousal, attention etc. remains unclear.

Importantly, specificity is not the central point of the present study. We concur with reviewer that perceptual, motor and autonomic responses to equally salient and threatening but non-painful stimuli might also be served by partially independent neural pathways. If so, this would indicate that the organizational principles revealed in the present study not only apply to pain but generalize to the processing of threat. Although we do not see how this can be experimentally addressed for the aforementioned reasons, we think that the possible lack of specificity potentially broadens the relevance of the present observations. Taken together, we therefore think that, in the particular context of the present study, control experiments on pain specificity are neither possible nor necessary.

We, however, fully agree on the interest of the topic and the lack of evidence for pain specificity in the present study. Based on the reviewer's suggestions and the considerations outlined above we have therefore substantially revised the manuscript to further clarify that pain specificity of the present observations remains unclear. In particular, we have changed the title as suggested. Moreover, we have added a paragraph to the discussion which summarizes the aforementioned considerations (p. 14, last paragraph). The added paragraph reads as follows.

“Some limitations apply to the present findings and their interpretation. ... Second, the specificity for pain remains unclear. fMRI (Mouraux et al., Neuroimage, 2011) and EEG (Mouraux et al., J Neurophysiol, 2009) studies have shown that most, if not all, brain responses to noxious stimuli are not pain-specific but rather reflect the salience of noxious events (Legrain et al., Prog Neurobiol, 2011). It is therefore likely that similar, partially independent patterns of neural responses mediate not only perceptual, motor and autonomic responses to noxious stimuli but also to equally salient and threatening stimuli from other modalities. However, the present findings provide direct evidence for partially independent processes serving perceptual, motor and autonomic responses only for noxious stimuli and the particular experimental conditions with eyes-closed.”

Further comments:

- P. 3: *The authors state "It has further been shown that gamma oscillations provide complementary information which can be particularly closely related to the perception of pain". Although it is true that several studies have shown a relationship between the perception of pain and gamma oscillations, these findings should be interpreted with caution. First, because gamma oscillations can also be observed in response to non-painful stimuli (e.g., Fardo et al. 2017, Journal of Neurophysiology). Second, because at least in some circumstances, the amplitude of pain-related gamma oscillations can be dissociated from pain perception (e.g., Liberati et al. 2018, Scientific Reports).*

We fully agree that gamma oscillations do not exclusively occur in response to noxious stimuli but are also observed in response to stimuli from other modalities. We further agree that gamma oscillations induced by noxious stimuli are often but not always (Liberati et al., Sci Rep, 2018; Tiemann et al., Pain, 2015) related to pain perception. We are therefore thankful for this suggestion and have now specified in the introduction (p. 3, second paragraph) and the discussion (p. 13, second paragraph) that the considerations in the manuscript refer to gamma oscillations in responses to noxious stimuli and that gamma oscillations are often but not always closely related to pain perception.

- P. 18: *How were the time window and the frequency range of gamma oscillations chosen? In particular, the 70-90 Hz frequency range seems very restricted, considering that any activity >40 Hz is generally regarded as gamma, and oscillations at frequencies higher than 90 Hz can normally be observed in response to sensory stimuli (e.g., see Zhang et al. 2012).*

The time window (150-350 ms) and frequency range (70-90 Hz) for the analysis of gamma oscillations was based on previous studies investigating gamma responses to noxious laser stimuli. In the manuscript, we have cited two studies (Tiemann et al., Pain, 2015; Schulz et al., J Neurophysiol, 2012) that showed maxima of gamma responses at these latencies and frequencies. We have carefully re-checked EEG (Hauck et al., Sci Rep, 2017; Tu et al., Hum Brain Mapp, 2016; Schulz et al., Cereb Cortex, 2015; Hauck et al., Front Hum Neurosci, 2015; Tiemann et al., Pain, 2015; Hu et al., Neuroimage, 2014; Valentini et al., Cortex, 2013; Schulz et al., Cereb Cortex, 2012; Schulz et al., Cereb Cortex, 2011), MEG (Hauck et al., Pain, 2013; Gross et al., Plos Biol, 2007) and intracranial recording (Liberati et al., Sci Rep, 2018; Liberati et al., Cereb Cortex, 2017; Liu et al., Neuroscience, 2015) studies investigating gamma responses to noxious laser stimuli. We found that nearly all studies found maxima of gamma responses in or close

to the time-frequency window chosen in the present study. Thus, the time-frequency window for the analysis of gamma oscillations is likely to capture gamma responses with an optimum signal-to-noise ratio which has now explicitly been stated in the methods section (p. 20, second paragraph). Moreover, we have added more references from different research groups and including larger number of subjects to further support this point.

- Gamma oscillations are quite difficult to observe on the scalp EEG, and a large number of subjects do not show a clear gamma response to thermonociceptive stimuli. In Fig. 3 only an average gamma response is shown. It would be important to understand how many participants out of 51 clearly showed this response, and how many were "non-responders". In particular, it would be important that the authors shared the data of the individual participants which have led to these results, e.g. on a public repository.

Figure 4 shows the individual time-frequency representations (TFRs) calculated on the concatenated data of all conditions for 50 participants. The TFRs show a discernible gamma response in the majority of subjects. This impression was confirmed by the results of dependent sample *t*-tests comparing gamma power in the predefined window (150-350 ms, 70-90 Hz) and the baseline period, calculated for every participant on single trial level, demonstrating a significant increase in gamma power in 34/50 participants, which equals a 68% rate of "gamma-responders".

Moreover, we appreciate the reviewer's suggestion to share the data of the individual participants on a public repository and fully agree that sharing data is important and worth supporting. In the meantime, we have therefore discussed the issue with our local ethics committee and the data security officer. Both have agreed that we can share the data on a public repository. We will thus upload the data to the *Open Science Framework*, www.osf.io, once the study has been published.

Figure 4. Individual ($n = 50$) time-frequency representations (TFRs) calculated on the concatenated data from all four conditions (perception, motor, autonomic, combined), displayed for the time period from $-0.4 - 1$ s and the frequency range from 30 - 100 Hz. Power values are scaled to the individual maximum.

- P. 18: *It is not clear why theta oscillations were not analyzed, and what the authors mean with "Responses at theta frequencies were not analyzed as they are known to mainly represent pain-evoked activity". Aren't other oscillations (e.g. gamma) also considered by the authors as pain-evoked activity?*

We thank the reviewer for the possibility to explain our approach more clearly. Brain activity at theta frequencies in response to noxious stimuli represents mainly phase-locked activity (Gross et al., 2007, PLoS Biol; Zhang et al., J Neurosci, 2012) and these phase-locked responses are well captured by the time domain analysis of laser-evoked potentials (May & Ploner 2018 Pain, Ploner et al 2017 TICS). We have clarified this point on page 20, last paragraph.

- P. 12: "The relationship between perceptual, motor and autonomic dimensions of pain is also relevant for the understanding of chronic pain". This statement is reasonable, but represents a large "leap" in the context of this study, in which only very brief noxious stimuli (1 ms) are used. These brief stimuli are predominantly mediated by quickly-adapting thermoreceptors. In contrast, the development of chronic pain has been related to slowly-adapting C fiber thermoreceptors (e.g., see Wooten et al. 2014, Nature Communications). It is therefore likely that the processing of transient nociceptive stimuli and the development of chronic pain underlie very different processes.

We concur with reviewer that the peripheral and central neural mechanisms underlying the processing of brief experimental pain stimuli likely differ from those underlying chronic pain. We have included this important consideration and supporting references (Ringkamp et al., Wall and Melzack's Textbook of Pain, 2013; Baliki and Apkarian, Neuron, 2015; Kuner and Flor, Nat Rev Neurosci, 2017) in the discussion section (p. 14, second paragraph).

- All stimuli were applied on the left hand, and motor responses were given using the right hand. How many participants were right/left handed? Wouldn't this aspect interfere with the reaction times?

All participants were right-handed (p.16, first paragraph). Right-handedness has been assessed by two key questions of handedness, i.e. which hand is used for writing and which hand is used for throwing a ball.

- The authors mention that the EEG was acquired while participants kept their eyes closed. This is likely to cause an alteration in the ongoing EEG oscillations, which should be taken into account (e.g. see Barry et al. 2007, Clinical Neurophysiology).

Eyes-open vs. eyes-closed states of participants have been shown to influence ongoing EEG oscillations and ongoing EEG oscillations can, in turn, influence the processing of pain (Tu et al., Hum Brain Mapp, 2016; Taesler and Rose, J Neurosci, 2016). We therefore agree that an indirect influence of eyes-open vs. eyes-closed states on brain responses to painful stimuli is possible. Consequently, the present findings obtained during eyes-closed do not necessarily generalize to eyes-open states. It is, however, important to note that the most important findings of the present study are similarities and differences across conditions which were all recorded during eyes-closed. On p. 15, first paragraph we have now explicitly stated that the present findings apply to the eyes-closed but not necessarily to the eyes-open state.

- P. 16: "Trials in which no or a delayed motor response occurred (reaction times > 650 ms) were discarded". Can the authors explain the reason for excluding these trials? Was it to avoid considering sensations mediated by C fibers?

Indeed, trials with reaction times > 650 ms were excluded to avoid considering sensations conducted by C-fibers. This is in accordance with previous studies (Mouraux et al 2003 Clin Neurophysiol, Mouraux et al 2011 J Neurosci). Moreover, the distribution of reaction times of the present study (Figure 5) indicates that this cutoff is appropriate to remove outliers and C-fiber mediated sensations. We have now added this explanation to the manuscript (page 18, first paragraph).

Figure 5. Distribution of reaction times in the motor condition.

- P. 17: How many trials were rejected due to artifact contamination?

We have now specified how many trials were rejected due to criteria applied to pain ratings and reaction times and how many were rejected due to artifact contamination (page 18, first paragraph and p.19, second paragraph). The paragraphs now read:

"After removing trials with ratings of 0 or 100, an average of 19 out of 20 trials per stimulus intensity (low: 18; medium: 20; high: 20) remained in the perceptual condition. After removing trials with missing reaction times and reaction times > 650 ms, an average of 18 out of 20 trials per stimulus intensity (low: 17; medium: 19; high: 19) remained in the motor condition."

"After removing trials contaminated by EEG artifacts, an average of 18 out of 20 trials per stimulus intensity remained in the perceptual (low: 18, range 11 - 20; medium: 18, range 13 - 20; high: 18, range 10 - 20) and the motor condition (low: 16, range 11 - 20; medium: 18, range 13 - 20; high: 18, range 10 - 20). In the autonomic condition, an average of 19 trials remained per stimulus intensity (low: 19, range 16 - 20; medium: 19, range 16 - 20; high: 19, range 16 - 20)."

- Fig. 1 would be clearer with a timeline showing the duration of the stimuli and the variable ISI.

We have improved Figure 1 according to the reviewer's suggestions.

- Fig. 2 would be more informative if the individual data were shown in a scatterplot. Bar graphs only showing average values are deceiving - especially considering that the standard deviations appear to be quite large.

We agree that it is informative to know more about the distribution of the data. To provide such information we have considered scatter plots, box plots and violin plots. We found that for the sample size of the present data a box plot yielded the best balance between information content and clarity. We have therefore replaced the bar graphs by box plots as shown below.

- The authors state that "The study was approved by the local ethics committee and conducted in conformity with the Declaration of Helsinki" (p. 14). According to the Declaration of Helsinki, "Every research study involving human subjects must be registered in a publicly accessible database before recruitment of the first subject". Can the Authors confirm that this is the case?

We thank the reviewer for bringing to our attention that the 2013 version of the Declaration of Helsinki includes that every research study in humans must be preregistered. To be honest, we have not been aware of this change to the previous version of the Declaration. In the 2008 version, preregistration had only been recommended for clinical trials. As the present study has not been preregistered, we have deleted the reference to the Declaration of Helsinki. Based on this important advice of the reviewer, the change in the policy of many scientific journals and the change of the NIH definition of clinical trials we have recently started to preregister all our experimental and clinical studies.

Reviewers' comments:

Reviewer #1 (Remarks to the Author):

This reviewer appreciates the effort of Authors to clear the points raised. The answers provided, however, did not clear all the concerns. As a result, the ms. gives a message which does not take into account the differences between the conditions in terms of the amplitudes of LEP components resulting from different instruction sets existing prior to the delivery of laser stimuli. In particular, the instruction set to elicit a speeded reaction time has increased the amplitude of the N1 component (best seen in the topographic plots of ICA-uncorrected data in Fig. 1 of the revision letter) which effect likely translated into the mediation weights causing an association between amplitudes of the N1 component and the shortening of reaction time in the motor condition. While Authors interpret the association as if the three conditions were equal and therefore the N1 component would be associated with the motor "dimension" of the task, the alternative interpretation would be that the motor response set has primed the cortical regions underlying the N1 component causing the mediation effect of the N1 component in the motor task. This interpretation differs from the suggested concept of a unitary pain experience having different "dimensions" each sub-served by different LEP components. A more valid concept would be that the context in which the noxious stimuli occur, defined by different instruction sets, manifests in different LEP components. The difference in the interpretations of results does not subtract from the value of the study.

It is suggested to provide a careful ANOVA analysis of amplitudes of the LEP parameters. The analysis provided in the revised ms. only states the presence of condition-related differences in N1 and gamma-band parameters, however, the reader does not know the directions of the condition effect. Fig. 1 of the revision letter shows the mean values of the parameters affected by the condition effect, however, the values have been transformed to Z-values, and no condition effect is seen. If the conditions differed significantly, the Z-transform applied across conditions has a questionable value as it only brushed off the differences which existed in the data. The ANOVA in page 6 of the ms. needs to be extended by showing the details of the condition effects on raw amplitudes of LEP components with a post-hoc analysis of the contrasts among the three conditions. Since individual mean values of LEP components (original, non-transformed data) are available, it is recommended to correlate the mediation weights in each condition with the raw amplitudes of the LEP components. This correlation or covariation analysis would say if the mediation effects could be related to the strength of the components in individual conditions.

The comparison of ICA-corrected and original topographic maps (Fig. 3, revision letter) shows that ICA has suppressed the spatial distribution of the N1 component. While reduction of eyeblink artefacts was a desirable effect of ICA, trimming of the temporally spreading N1 component was an undesirable effect, and an effect which could have skewed the results. While it is too late to ask for a change at this stage, it seems that a regression approach toward elimination of eyeblink artefacts (e.g., the pattern matching algorithm) would have been more prudent in this data set. The N1 topographic maps presented differ from the recent modelling article (Bradley et al., Hum. Brain Map., 2017, Fig. 1) and a large number of previous articles which have showed a negative potential in the contralateral and even ipsilateral temporal electrodes at the latency of the N1 potential (about 170 ms).

As far as the quantification of N1 component is concerned, the N1 component at T8 electrode would be much less affected by the rising N2 component seen at electrode Cz than the electrode C4. The N1 component as seen in Fig. 2 of the revision letter occurred earlier at electrode T8 than C4 and it is also discernible from the N2 potential (not so in C4-Fz data). Therefore, the time interval selected to quantify the N1 component would need to be moved some milliseconds towards the shorter latency.

Authors addressed the concern about their interpretation of scalp potential data using anatomical

labels of underlying cortical generators by adding one sentence in page 12. Either one dismisses the source localisation approach and refrains to solely describing scalp data or employs with appropriate caution some source localisation method and has then the right to discuss the possible generators. Source localisation depends on a number of conditions such as electrode layouts, number of electrodes, noise level, pre-processing steps (e.g., filtering, ICA), or the lead field parameters and therefore, it is not known which sources sub-served individual LEP components in the present study in each of three different conditions. It is recommended to either add a source dipole localisation section to the Results or to avoid in Discussion any references to the cortical generators as being "likely" involved. As it stands, the sentence "When discussing the potential generators of EEG responses, it is important to bear in mind that evidence on the neural generators of LEP using source reconstruction is inherently ambiguous." (p.12) has only served to justify the interpretation of findings using source modelling data retrieved from previous studies, which does not sound correct.

Minor: It is recommended to reconsider the term "pain dimensions" which is frequently used in the ms. (including Abstract). Pain is an unpleasant subjective experience having a sensory, cognitive-evaluative, and affective dimensions (Melzack and Casey, 1968). The ms. did not study these traditional dimensions of pain, rather it analysed selected responses to noxious laser stimuli.

Reviewer #2 (Remarks to the Author):

The authors should be commended for their comprehensive response to reviews and willingness to include additional data and analyses. The inclusion of the combined task supports their original findings, and they softened language and discussed additional work where appropriate. I think this paper will make an important contribution to the field and recommend acceptance.

Reviewer #3:

[This reviewer did not leave any comments for the authors, but in their comments to the editors, they indicated that the authors have addressed all of the earlier issues, and suggests that Figure 4 of the rebuttal is very informative and that it should be included in the Suppl. Mat.]

Response to referees

Reviewer #1

This reviewer appreciates the effort of Authors to clear the points raised. The answers provided, however, did not clear all the concerns. As a result, the ms. gives a message which does not take into account the differences between the conditions in terms of the amplitudes of LEP components resulting from different instruction sets existing prior to the delivery of laser stimuli. In particular, the instruction set to elicit a speeded reaction time has increased the amplitude of the N1 component (best seen in the topographic plots of ICA-uncorrected data in Fig. 1 of the revision letter) which effect likely translated into the mediation weights causing an association between amplitudes of the N1 component and the shortening of reaction time in the motor condition. While Authors interpret the association as if the three conditions were equal and therefore the N1 component would be associated with the motor “dimension” of the task, the alternative interpretation would be that the motor response set has primed the cortical regions underlying the N1 component causing the mediation effect of the N1 component in the motor task. This interpretation differs from the suggested concept of a unitary pain experience having different “dimensions” each sub-served by different LEP components. A more valid concept would be that the context in which the noxious stimuli occur, defined by different instruction sets, manifests in different LEP components. The difference in the interpretations of results does not subtract from the value of the study.

We thank the referee again for carefully reviewing our manuscript and his/her overall positive assessment of the study. We carefully considered the current comments and agree that task effects on amplitudes of brain responses are an interesting and relevant topic. In the present study, such task effects on amplitudes of brain responses were observed for the N1 and gamma responses. We have therefore performed correlation analyses between mediation effects and amplitudes, as suggested (see our response to the next point). The results did not show any significant correlation between mediation weights and amplitudes (Table 2). Moreover, the applied z-transformation of brain responses across participants and trials for each condition controls for amplitude differences between conditions. The differences of mediation effects between conditions can therefore not be influenced by amplitude differences. These findings and procedures reliably control for amplitude effects on the results of the mediation analyses. This has been explicitly discussed on p. 14, last paragraph of the manuscript.

It is suggested to provide a careful ANOVA analysis of amplitudes of the LEP parameters. The analysis provided in the revised ms. only states the presence of condition-related differences in N1 and gamma-band parameters, however, the reader does not know the directions of the condition effect. Fig. 1 of the revision letter shows the mean values of the parameters affected by the condition effect, however, the values have been transformed to Z-values, and no condition effect is seen. If the conditions differed significantly, the Z-transform applied across conditions has a questionable value as it only brushed off the differences which existed in the data. The ANOVA in page 6 of the ms. needs to be extended by showing the details of the condition effects on raw amplitudes of LEP components with a post-hoc analysis of the contrasts among the three conditions. Since individual mean values of LEP components (original, non-transformed data) are available, it is recommended to correlate the mediation weights in each condition with the raw amplitudes of the LEP components. This correlation or

covariation analysis would say if the mediation effects could be related to the strength of the components in individual conditions.

We have now performed post-hoc tests comparing the N1 and gamma amplitudes across conditions, as suggested (Table 1). The results show that N1 amplitudes were higher in the *motor* than in the *perception* and *autonomic* conditions. Gamma amplitudes were higher in the *motor* than in the *autonomic* conditions. We have added these findings to the manuscript on p. 6, last paragraph.

	Condition [mean \pm SD]			Post hoc tests [t (p)]		
	perception	motor	autonomic	p vs. m	m vs. a	p vs. a
N1	-3.11 \pm 2.44	-4.57 \pm 2.65	-3.13 \pm 2.17	4.37 (<0.001)	-4.29 (<0.001)	0.09 (1)
γ	0.004 \pm 0.009	0.009 \pm 0.02	0.003 \pm 0.007	-2.53 (0.06)	3.02 (0.01)	1.31 (0.6)

Table 1 Amplitudes of brain responses per condition and post hoc pairwise comparisons. Given are the amplitudes (N1, μ V; gamma, μ V²) and standard deviations of the N1 and gamma responses in all three conditions as well as the results of post hoc pairwise comparisons. All p -values are Bonferroni-corrected. Significant effects are marked in red. *p*, perception; *m*, motor; *a*, autonomic; SD, standard deviation.

Moreover, we have correlated the mediation weights with the raw amplitudes of brain responses, as suggested (Table 2). The results do not show a significant correlation for any brain responses in any condition. These findings further confirm that response amplitudes cannot explain the observed mediation effects. As we feel that the manuscript is already quite complex we would prefer not to add these correlations to the manuscript.

	Correlation coefficients [Pearson's r (p)]		
	perception	motor	autonomic
N1	-0.23 (0.11)	0.00 (0.98)	-0.12 (0.50)
N2	-0.27 (0.06)	-0.15 (0.31)	-0.25 (0.17)
P2	0.16 (0.25)	-0.07 (0.61)	0.01 (0.95)
γ	0.07 (0.63)	0.06 (0.69)	0.07 (0.68)

Table 2 Pearson correlations between mediation weights and amplitudes of brain responses. Given are the correlation coefficients between the single-subject mediation weights and the baseline-corrected, non-z-standardized amplitudes of brain responses. No significant correlations were found. All p -values are uncorrected.

The comparison of ICA-corrected and original topographic maps (Fig. 3, revision letter) shows that ICA has suppressed the spatial distribution of the N1 component. While reduction of eyeblink artefacts was a desirable effect of ICA, trimming of the temporally spreading N1 component was an undesirable effect, and an effect which could have skewed the results. While it is too late to ask for a change at this stage, it seems that a regression approach toward elimination of eyeblink artefacts (e.g., the pattern matching algorithm) would have been more prudent in this data set. The N1 topographic maps presented differ from the recent modelling

article (Bradley et al., Hum. Brain Map., 2017, Fig. 1) and a large number of previous articles which have showed a negative potential in the contralateral and even ipsilateral temporal electrodes at the latency of the N1 potential (about 170 ms).

Amplitudes of N1 responses were lower after ICA-correction which likely reflects the intended removal of artifacts. Artifact removal of EEG data is in general a difficult and controversial topic and many different approaches have been proposed including regression-based approaches. Regression-based approaches are likely highly effective in removing eyeblink artifacts but are less likely to control for muscle artifacts as these are less monomorphic than eyeblink artifacts. We have therefore deliberately decided to apply a well-established ICA-based approach.

As far as the quantification of N1 component is concerned, the N1 component at T8 electrode would be much less affected by the rising N2 component seen at electrode Cz than the electrode C4. The N1 component as seen in Fig. 2 of the revision letter occurred earlier at electrode T8 than C4 and it is also discernible from the N2 potential (not so in C4-Fz data). Therefore, the time interval selected to quantify the N1 component would need to be moved some milliseconds towards the shorter latency.

The use of the C4-Fz electrode montage for analyzing the N1 response is in accordance with previous findings and recommendations (Hu et al., Neuroimage, 2010). Moreover, the definition of the time window was motivated by previous studies analyzing the N1 response (e.g. Hu et al., Neuroimage, 2010; Moayed et al., Cereb Cortex, 2015; Tiemann et al., Pain, 2015). It might appear appropriate to adjust the N1 analysis time window to the present data. However, the selection of the time window based on the same data which were analyzed would imply circularity and eventually result in invalid statistical inferences (Kriegeskorte et al., Nat Neurosci, 2009). We therefore believe that it is appropriate to stick to the predefined time window.

Authors addressed the concern about their interpretation of scalp potential data using anatomical labels of underlying cortical generators by adding one sentence in page 12. Either one dismisses the source localisation approach and refrains to solely describing scalp data or employs with appropriate caution some source localisation method and has then the right to discuss the possible generators. Source localisation depends on a number of conditions such as electrode layouts, number of electrodes, noise level, pre-processing steps (e.g., filtering, ICA), or the lead field parameters and therefore, it is not known which sources sub-served individual LEP components in the present study in each of three different conditions. It is recommended to either add a source dipole localisation section to the Results or to avoid in Discussion any references to the cortical generators as being “likely” involved. As it stands, the sentence “When discussing the potential generators of EEG responses, it is important to bear in mind that evidence on the neural generators of LEP using source reconstruction is inherently ambiguous.” (p.12) has only served to justify the interpretation of findings using source modelling data retrieved from previous studies, which does not sound correct.

With respect to the discussion of the generators of brain responses to noxious stimuli, we respectfully disagree. Although we are well aware that different conditions can significantly influence source localization results, we feel that the synopsis of source localization studies based on scalp recordings (e.g. Garcia-Larrea et al., Neurophysiol Clin, 2003) together with intracerebral recordings (e.g. Bradley et al., Hum Brain Mapp,

2017) provide enough information on the generators of responses to discuss them with an appropriate caveat as done on p. 12, last paragraph of the manuscript. We would therefore kindly prefer to keep the considerations of the generators of responses in the discussion.

Minor: It is recommended to reconsider the term “pain dimensions” which is frequently used in the ms. (including Abstract). Pain is an unpleasant subjective experience having a sensory, cognitive-evaluative, and affective dimensions (Melzack and Casey, 1968). The ms. did not study these traditional dimensions of pain, rather it analysed selected responses to noxious laser stimuli.

We thank the reviewer for this suggestion and re-considered the use of the term *dimension* in the manuscript. We understand that our use of the term *pain dimensions* does not fully match the *determinants of pain* described by Melzack and Casey (1968). We, however, feel that the use of the term *dimension* in the present study is sufficiently clear to allow the reader for understanding what is meant. We would therefore kindly prefer to retain the term *dimension*.

Reviewer #2

The authors should be commended for their comprehensive response to reviews and willingness to include additional data and analyses. The inclusion of the combined task supports their original findings, and they softened language and discussed additional work where appropriate. I think this paper will make an important contribution to the field and recommend acceptance.

We are very grateful for these positive and encouraging comments.

Reviewer #3

[This reviewer did not leave any comments for the authors, but in their comments to the editors, they indicated that the authors have addressed all of the earlier issues, and suggests that Figure 4 of the rebuttal is very informative and that it should be included in the Suppl. Mat.]

We have included Fig. 4 from the previous reply letter as Suppl. Fig. 3 to the manuscript, as suggested.

REVIEWERS' COMMENTS:

Reviewer #1 (Remarks to the Author):

Thank you for addressing the comments of concern.